# A single cluster of RNA Polymerase II molecules is stably associated with active genes

Apratim Mukherjee [1,2], Manya Kapoor[2,3,9], Kareena Shankta[2,4,9], Samantha Fallacaro [2,5], Raymond D. Carter [2,6], Gabriela Hayward-Lara[2,5], Puttachai Ratchasanmuang[2,7], Yara I. Haloush[2] & Mustafa Mir [1,2,7,8] ✉

In eukaryotic nuclei, transcription is associated with the clustering of RNA Polymerase II (RNAPII) molecules. The mechanisms underlying cluster formation, their interactions with genes, and their impact on transcriptional activity remain heavily debated. Here, we take advantage of the naturally occurring increase in transcriptional activity during Zygotic Genome Activation (ZGA) in *Drosophila melanogaster* embryos to characterize the functional roles of RNAPII clusters in a developmental context. Using single-molecule tracking and lattice light-sheet microscopy, we find that RNAPII cluster formation depends on transcription initiation and that cluster lifetimes depend on transcriptional activity when not constrained by interphase duration. We show that single clusters are stably associated with active gene loci during transcription and that cluster intensities are strongly correlated with transcriptional output. Collectively, our data and simulations on cluster formation kinetics show that RNAPII clusters reflect local accumulations of transcriptionally engaged polymerases and do not form through higher-order mechanisms such as phase separation.

It has long been hypothesized that the organization of the nucleus into enzymatically active sub-compartments that concentrate reactants may boost the efficiency of biochemical reactions[1–4]. In the context of gene expression, this idea gained support from observations in the early 1990s that suggested RNA synthesis and processing occur within discrete nucleoplasmic foci[5,6]. These foci, termed transcription factories, were thought to be stable assemblies containing 10s-100s of RNAPII molecules associated with multiple transcriptionally active genes[7,8]. Super-resolution imaging in fixed cells challenged this model and suggested that, on average, transcriptional foci represent single

RNAPII molecules[9]. Also in contrast to the transcription factory model, RNA fluorescence in situ hybridization imaging in fixed *Drosophila* embryos suggested that each active gene is associated with its own RNAPII cluster[10]. On the other hand, live-cell single-molecule imaging has shown that RNAPII clusters composed of multiple molecules are prevalent but are highly dynamic, lasting on the order of just seconds[11]. These live-cell approaches suggested that cluster formation is tightly linked to transcription initiation and that cluster lifetimes determine the number of polymerases loaded onto a gene[11,12]. In contrast, recent single-molecule tracking of RNAPII suggests that clusters form post-

[1]Department of Cell and Developmental Biology, Perelman School of Medicine, University of Pennsylvania, Philadelphia, PA, USA. [2]Center for Computational and Genomic Medicine, Children's Hospital of Philadelphia, Philadelphia, PA, USA. [3]Department of Bioengineering, University of Pennsylvania, Philadelphia, PA, USA. [4]Roy and Diana Vagelos Program in Life Sciences and Management, University of Pennsylvania, Philadelphia, PA, USA. [5]Developmental, Stem Cell, and Regenerative Biology Graduate Group, Perelman School of Medicine, Philadelphia, PA, USA. [6]Biochemistry, Biophysics, and Chemical Biology Graduate Group, Perelman School of Medicine, University of Pennsylvania, Philadelphia, PA, USA. [7]Howard Hughes Medical Institute, Children's Hospital of Philadelphia, Philadelphia, PA, USA. [8]Epigenetics Institute, University of Pennsylvania Perelman School of Medicine, Philadelphia, PA, USA. [9]These authors contributed equally: Manya Kapoor, Kareena Shankta. ✉e-mail: mirm@chop.edu

initiation and are composed of ~6-7 elongating polymerases at a gene[13]. Given these contrasting observations made using high resolution imaging methods, it is still debated whether clusters form at initiation and dissolve prior to elongation, or only form post-initiation and represent a collection of elongating molecules at a single gene.

More recently, larger clusters of RNAPII and co-activators described as condensates, spanning half a micrometer and lasting tens of minutes, have also been observed[14,15]. These condensates were initially proposed to form at super-enhancers through liquid-liquid phase-separation and transiently interact with target genes to up-regulate transcription[14]. However, subsequent studies demonstrated that their formation may not depend on super-enhancers, and that transcriptional upregulation requires condensate to gene proximities of less than a micrometer rather than direct contact[16]. Recent imaging in *Drosophila* embryos also suggests that RNAPII clusters transiently associate with active transcription sites and dissipate as transcription increases[17] consistent with models in which local RNA concentrations promote condensate formation at low levels and lead to their dissipation at higher concentrations[18]. The transient associations of these larger RNAPII condensates with genes contrast with the findings on more dynamic clusters described above. As these larger clusters have been reported to represent ~10% of the RNAPII clusters in the nuclei in which they were identified[14], their interactions with active genes may

not be representative of the majority of clusters. The nature of RNAPII cluster interactions with actively transcribing genes may thus be multifaceted and remains unclear.

Here we examine how RNAPII clustering changes in response to the sharp increase in transcriptional activity that occurs during Zygotic Genome Activation (ZGA) in *Drosophila melanogaster* embryos. Using live-embryo single-molecule tracking, lattice light-sheet microscopy, imaging of transcriptional activity, small molecule perturbations to transcription, and simulations, we address mechanisms of cluster formation, changes in biophysical properties during transcriptional upregulation, and how clusters interact with active genes. We find that RNAPII cluster formation depends on transcription initiation and that their lifetimes depend on transcriptional activity. RNAPII clusters transition from being enriched in initiating polymerases prior to ZGA to being composed primarily of elongating molecules after ZGA. Furthermore, we find that individual RNAPII clusters persistently associate with a single active transcription site throughout the duration of a transcription burst.

## Results

### Chromatin-bound fraction of RNAPII molecules increases during Zygotic Genome Activation

The major wave of ZGA in *Drosophila* embryos occurs in the 14th nuclear cleavage cycle (nc14) during which the amount of transcription increases sharply along with the number of active genes. There are ~1,000 active genes before nc14, increasing to upwards of 3,500 genes in nc14, with estimates varying by the method of quantification[19–23] (Fig. 1a). We reasoned that this increase in transcriptional activity should correspond to an increase in the chromatin-bound fraction of RNAPII molecules when measured using single-molecule tracking. To quantify changes in RNAPII molecular kinetics we endogenously tagged RPB1, the largest and catalytic subunit of RNAPII, at its N-terminus, with the photoconvertible fluorescent protein mEos3.2 (Fig. S1. 1). This insertion is homozygous-viable over hundreds of generations of propagation and only homozygous embryos were imaged. We performed fast single-molecule tracking (10 msec/frame) (Fig. 1b and Fig. S1. 2a, Movie S1)[24–26]. We used state-array modeling to infer diffusion coefficients[27] for each trajectory, and obtained distributions of diffusion coefficients for the histone H2B and nuclear localization sequence (NLS) fused to mEos3.2 in addition to RNAPII. Based on the diffusion coefficient distributions of H2B and NLS, we categorized RNAPII trajectories as (i) chromatin bound; (ii) intermediate, which may reflect a heterogeneous population of molecules including those transiently confined, or diffusing in complex with co-factors, or (iii) fast, which may reflect freely diffusing molecules or truncated products with reduced molecular weights (Fig. 1c and Fig. S1. 1b). We find that the fraction of chromatin-bound RNAPII molecules increases from 37 ± 5% in nc13, to 51 ± 8% in nc14, representing a 1.4 ± 0.3-fold increase (Fig. 1d and Fig. S1. 2c). This increase in the bound population is accompanied by a concomitant 1.6 ± 0.4-fold decrease in the fast population and 1.2 ± 0.4-fold decrease in the intermediate population. Overall, RNAPII molecules transition, from fast and intermediate kinetic states, to bound, during ZGA, consistent with increasing transcriptional activity. However, our single-molecule localization error of ~30 nm is insufficient to distinguish between RNAPII molecules engaged in different steps of the transcription cycle, namely engagement, initiation, and elongation, all of which would appear as chromatin-bound in our data. To deconvolve this chromatin-bound population, we next turned to small-molecule inhibitors that selectively block specific stages of the transcription cycle[28].

### Transcription initiation and elongation differentially contribute to the RNAPII bound fraction during ZGA

To assess the relative contributions of molecules engaged in transcription initiation and elongation to the bound fraction of

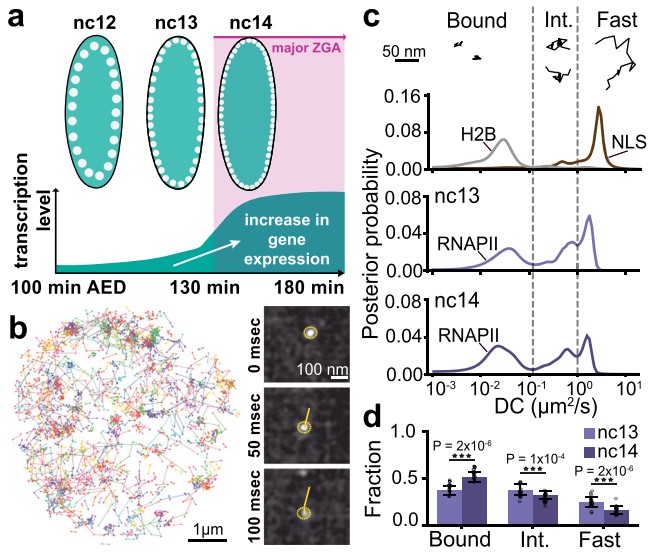

**Fig. 1 | Single molecule kinetics of RNAPII during Zygotic Genome Activation (ZGA). a** Timeline of zygotic genome activation (ZGA) in *Drosophila* embryos from nuclear cycles (nc) 12-14. x-axis shows minutes after egg deposition (AED). **b** (Left) Representative nucleus showing individual mEos3.2-RPB1 single molecule tracks over 80 s acquired at 10 ms exposure time. (Right) Timelapse snapshots showing a single molecule (yellow circle) being tracked (yellow line) from the same nucleus over 10 frames (100 msec). **c** Representative individual trajectories within the different kinetic bins (bound, intermediate and fast) for RPB1. Diffusion coefficient distributions for H2B and NLS (top) in nc14, RPB1 in nc13 (middle panel), and in nc14 (bottom panel). Dashed vertical lines show the division of the diffusion coefficient distributions into kinetic bins for RPB1. A total of 210,638 trajectories from 9 embryos (270 nuclei) were obtained in nc13, and 205,532 trajectories from 10 embryos (264 nuclei) in nc14, for RPB1. A total of 37,856 trajectories from 3 embryos (84 nuclei) and 8,009 trajectories from 3 embryos (76 nuclei) were obtained for H2B and NLS respectively in nc14. **d** Bound, intermediate, and fast fractions for RPB1 in nc13 and nc14. Bars show the mean across fields of view. Each data point represents a bound fraction value obtained from 23 single fields of views ( ~8–15 nuclei) and error bars show standard deviations. Two sided Mann-Whitney U-test was performed to determine significance and following *p*-values were used: *p < 0.05, **p < 0.01, and ***p < 0.001. Source data are provided as a Source Data file.

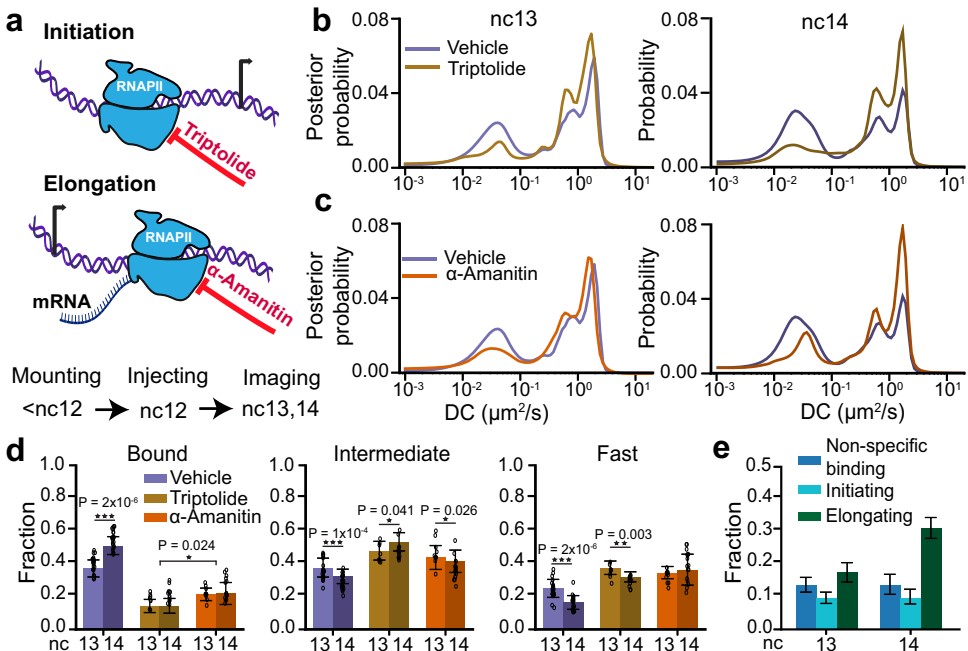

**Fig. 2 | Changes in RNAPII kinetics upon inhibition of transcriptional elongation or initiation. a** Schematic of transcription cycle stages targeted by triptolide, to inhibit initiation, and α-amanitin, to inhibit elongation, and overview of experimental timeline. **b** RPB1 diffusion coefficient distributions in vehicle and triptolide injected embryos in nc13 (left) and nc14 (right). **c** RPB1 diffusion coefficient distributions in vehicle and α-amanitin injected embryos in nc13 (left) and nc14 (right). 210,638 trajectories from 9 embryos (270 nuclei) in nc13 and 205,532 trajectories from 10 embryos (264 nuclei) in nc14 were obtained for vehicle injected embryos; 40,775 trajectories from 3 embryos (70 nuclei) in nc13 and 57,003 trajectories from 5 embryos (84 nuclei) in nc14 were obtained for triptolide injected embryos; 37,885 trajectories from 5 embryos (74 nuclei) in nc13 and 55,474 trajectories from 6 embryos (82 nuclei) in nc14 were obtained for α-amanitin injected embryos.

**d** Comparison of bound (left), intermediate (center), and fast (right) fractions for RPB1 in nc13 and nc14 for all conditions. Data points represent 23 individual fields of view for vehicle-injected embryos in both nc13 and nc14, 9 and 14 individual fields of view for triptolide injected embryos in nc13 and nc14 respectively, and 11 and 16 individual fields of view for α-amanitin injected embryos in nc13 and nc14 respectively with ~8-15 nuclei per field of view. Bars show the mean across the fields of view. All error bars show standard deviations. Two sided Mann-Whitney U-test was performed to determine significance and following p-values were used: *$p < 0.05$, **$p < 0.01$, and ***$p < 0.001$. **e** RNAPII bound fraction split into non-specific binding, initiating, and elongating in nc13 and 14. Bars show point estimates of each component computed from the mean bound fractions. Error bars represent standard propagation of error. Source data are provided as a Source Data file.

RNAPII, we injected embryos with either triptolide, which blocks formation of the transcription initiation complex by targeting TFIIH[29], or α-amanitin, which binds to RNAPII directly and slows the elongation rate to an extent that transcription is effectively inhibited[30,31] (Fig. 2a). We established an effective working concentration of triptolide by performing dosage analyses (Fig. S2. 1b). Concentrations above 1.75 mg/ml caused immediate developmental arrest or defects, whereas doses between 0.5 and 1.75 mg/ml allowed embryos to survive until at least the middle of nc14 without detectable abnormalities. We selected the 0.5 mg/ml dose as the working concentration and used EUTP labelling to confirm the loss of nascent transcripts (Fig. S2. 1c). The concentration of α-amanitin was selected at 0.5 mg/ml based on previous reports[32]. Using the MS2–MCP reporter system to mark sites of transcription of hunchback (*hb*, an actively transcribed gene during this stage of embryogenesis), and lattice light-sheet imaging, we found that injection of either inhibitor at these doses during nc12 completely ablates detectable MS2 transcription within 5 minutes while embryos continued to develop until mid-nc14, after which they failed to gastrulate (Fig. S2.1d, e; Supplementary Movies S2–4). In contrast, vehicle-injected embryos retained MS2 foci and developed normally through gastrulation, confirming that the injection procedure itself does not impair viability (Fig. S2. 1d, e). Furthermore, we measured the mean nuclear intensity of RNAPII across vehicle, triptolide, and α-amanitin injected embryos in nc13 and nc14 and found no significant differences suggesting that neither injection led to global degradation of RNAPII over the time course of our imaging

experiments (Fig. S2.1g). Finally, the bound, intermediate, and fast fractions of RNAPII do not change between uninjected and vehicle-injected embryos, demonstrating that vehicle injection itself does not significantly alter RNAPII activity (Fig. S2. 1h, i).

Inhibiting transcription initiation causes a significant reduction in the bound fraction of RNAPII, with a $2.6 \pm 0.7$-fold decrease in nc13 and a $3.6 \pm 1.0$-fold decrease in nc14 compared to control embryos (Fig. 2b, d and Fig. S2. 1F). This decrease in the bound fraction is consistent with triptolide's mechanism of action, which blocks pre-initiation complex (PIC) formation by binding to the XPB helicase subunit of TFIIH, a key general transcription factor (GTF)[33]. This inhibition reduces the pool of RNAPII available for chromatin engagement. In both nc13 and nc14, $14 \pm 5\%$ of molecules remain bound after inhibiting initiation. We interpret this residual bound fraction as non-specific RNAPII-chromatin interactions though we cannot exclude the possibility that some molecules are not evicted by triptolide. In comparison, treatment by α-amanitin decreased the bound fraction by $1.7 \pm 0.4$-fold in nc13 and $2.3 \pm 0.8$-fold in nc14 compared to control embryos (Figs. 2c, d, S2.1f). We reasoned that the residual bound fraction after inhibiting elongation reflects RNAPII molecules that have initiated transcription as well as those engaged in non-specific interactions. This is consistent with previous studies which suggest that α-amanitin traps RNAPII in an open inactive conformation by interfering with the trigger loop and bridge helix, causing it to cycle between initiation and premature eviction from chromatin[30,34,35]. Thus, the bound fractions from triptolide and α-amanitin injected embryos can be used to estimate the fractions of initiating and non-specifically interacting molecules within the bound fraction in vehicle injected embryos. Using this fact we

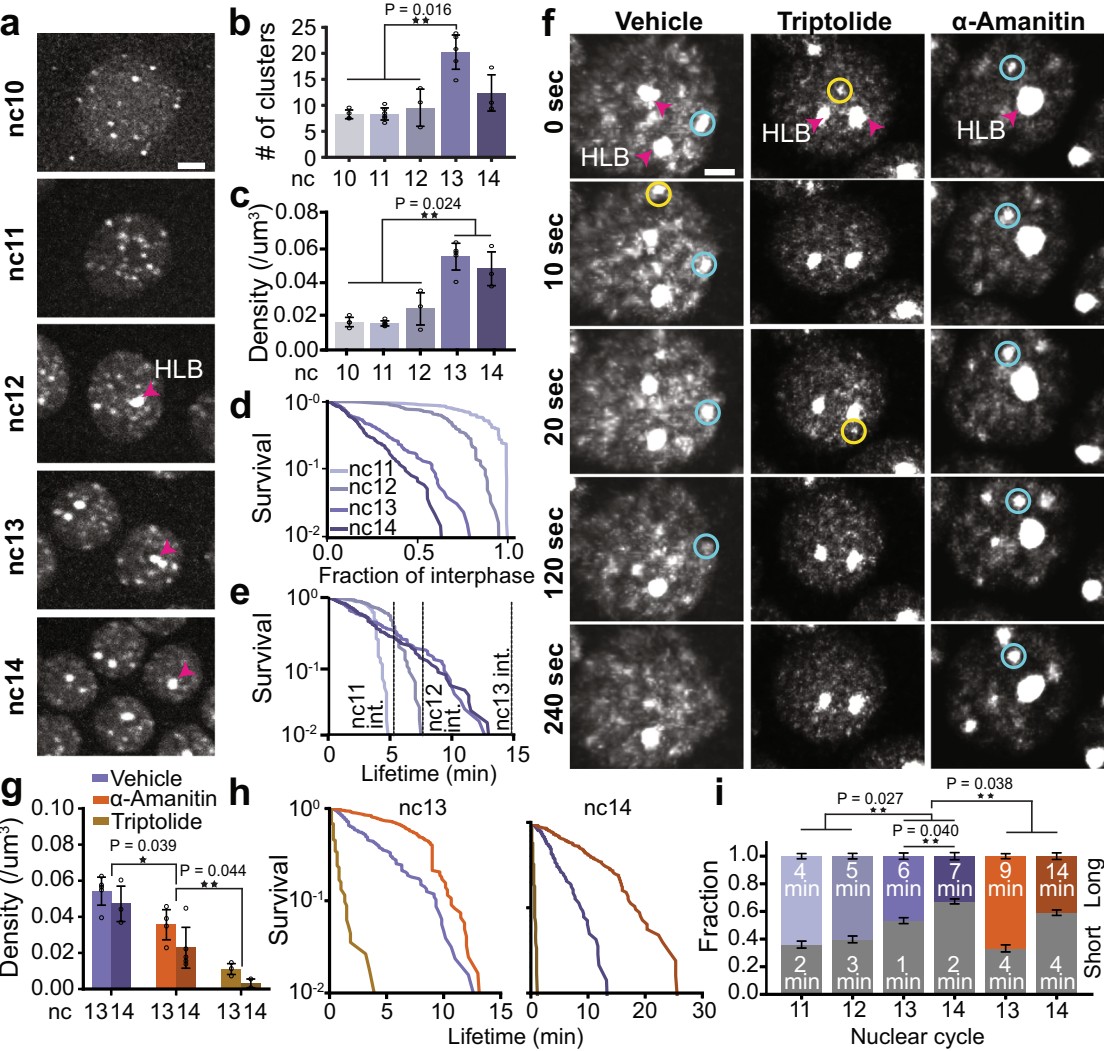

**Fig. 3 | RNAPII cluster kinetics through Zygotic Genome Activation and transcriptional inhibition. a** Deconvolved maximum-intensity projection images of eGFP-RPB1 in nuclei from nc 10–14. Pink arrow indicates histone locus body (HLB). The scale bar is 6 µm. **b, c** Quantification of cluster number and density (clusters/ µm³) from nc 10-14. Each point is an individual embryo; bars show mean across the embryos; error bars show standard deviation. $n = 4, 6, 3, 5$, and 3 embryos for nc 10–14 respectively. Two sided Mann-Whitney U-test was performed to determine significance and following p-values were used: *$p < 0.05$, **$p < 0.01$, and ***$p < 0.001$. **d** Survival curves of cluster lifetimes normalized to the interphase duration for nc11-14. The survival curve for nc10 is not shown because clusters last the entire interphase. **e** Absolute cluster lifetimes for nc11-14. 120 clusters measured per cycle from at least 3 embryos with at least 5 nuclei per embryo. **f** Time series of eGFP-RPB1 nuclei from vehicle, triptolide, and α-amanitin injected embryos. Intensities are scaled identically. Yellow and blue circles are examples of short- and long-lived clusters respectively. Pink arrows indicate HLBs. The scale bar is 2 µm. **g** Cluster density in vehicle, triptolide, and α-amanitin injected embryos in nc 13 and 14. Data points are individual embryos. $n = 5, 3$ for the control embryos in nc 13, 14 respectively, $n = 3$ for both triptolide and α-amanitin injected embryos in nc 13,14.

Bars show mean across the embryos. Error bars represent standard deviation. Two sided Mann-Whitney U-test was performed to determine significance and following p-values were used: *$p < 0.05$, **$p < 0.01$, and ***$p < 0.001$. **h** Cluster survival probability for vehicle, triptolide, and α-amanitin in nc 13 and 14. 120 clusters were measured for each condition and nuclear cycle over at least 3 independent movies and at least 5 nuclei per movie. **i** Cluster fractions from a 2-state Gaussian Mixture Model (GMM) fit. Stacked bars show the GMM weights of short-lived clusters (in grey, bottom) and the long-lived clusters (in color, top) for vehicle and α-amanitin embryos across different nuclear cycles. 120 clusters were measured per cycle from at least 3 embryos with at least 5 nuclei per embryo. Text inside each bar shows the mean lifetime of clusters for that condition (nuclear cycle and treatment) and category (short, long) in minutes. Error bars represent the uncertainty of the mixture weights using the number of clusters fit in that condition. Triptolide is not shown due to low cluster counts and no long-lived population. Two sided Mann-Whitney U-test was performed to determine significance and following p-values were used: *$p < 0.05$, **$p < 0.01$, and ***$p < 0.001$. Source data are provided as a Source Data file.

decomposed the bound fraction in vehicle injected embryos into initiating, elongating, and non-specifically interacting molecule components (Fig. 2e). This decomposition shows that the elongating fraction of RNAPII increases by ~1.8 ± 0.5-fold from nc13 to nc14. Prior work suggests that the properties of RNAPII clusters, including their numbers and lifetimes, depend on the relative levels of transcriptional initiation and elongation. This prompted us to next examine our single molecule results in the context of changes in RNAPII clustering during ZGA[12,14,36].

## Transcription initiation drives cluster formation while elongation destabilizes them

To examine RNAPII clustering during ZGA, we performed volumetric lattice light-sheet imaging in *Drosophila* embryos with endogenous RPB1 homozygously tagged with eGFP at its N-terminus at both alleles (Figure. S1. 1). We found that RNAPII forms distinct clusters as early as nc10 while the more prominent Histone Locus Bodies (HLBs) first emerge at nc12 (Fig. 3a, Figure. S3. 1, and Movie S5). The HLBs are easily identified as the two largest and brightest RNAPII bodies in each

nucleus and form exclusively at clusters of replication dependent histone genes which contain ~100 copies of each histone gene[37]. Quantification of the detectable cluster numbers, excluding HLBs, shows that the average number of clusters per nucleus is initially stable from nc10-12 at ~8 per nucleus, and sharply increases by ~2-fold in nc13. Cluster number then decreases down to ~12 per nucleus in nc14 (Fig. 3b, Figure S3. 2a–c). Although the nuclear volume decreases by ~29% from nc 10 to 12, the cluster density (number of clusters per μm³) remains relatively stable during this period[38]. However, in nc13, the ~2-fold increase in cluster number combined with a 46% decrease in nuclear volume results in a pronounced increase in cluster density. Notably, as the nuclear volume continues to decrease further by ~37% from nc13 to nc14[38], the cluster density remains relatively constant (Fig. 3c).

Next, we investigated whether changes in cluster number and density were accompanied by changes in cluster lifetimes. Since interphase times increase significantly during *Drosophila* embryogenesis from nc11 (~6–7 min) to nc14 (~25 min prior to cellularization), we first normalized individual cluster lifetimes to their respective interphase lengths[39]. While the majority of clusters in the early cycles (nc11-12) persist through interphase, we found that cluster lifetimes, when normalized to interphase duration, shorten significantly as ZGA progresses (Fig. 3d). This reduction in lifetimes suggests that in nc11 and nc12, absolute cluster lifetimes (Fig. 3e) are constrained by the short interphase durations consistent with previous reports of frequently aborted transcription during these stages[19]. In contrast, during later cycles (nc13-14), absolute cluster lifetimes remain similar but are no longer capped by interphase duration, leading to shorter normalized values (Fig. 3d, e). The distribution of cluster lifetimes are well described by a short and long-lived population (Fig. S3. 2d, Movie S8). Strikingly, the fraction of long-lived clusters decreases from 64 ± 2% in nc11 down to only 33 ± 3% in nc14. The data suggest that cluster lifetime in nc13 and nc14, may not be limited by cycle length but rather depends on the rate of transcriptional activity, as supported by the concurrent rise in the elongating fraction of RNAPII (Fig. 2e). Consistent with this shift in lifetimes, immunofluorescence analysis shows that RNAPII molecules in clusters shift from being in mostly initiating states (Ser5P staining) during the early cycles (nc11-12), to largely elongating states (Ser2P staining) in nc14 (Fig. S3. 3).

Upon inhibiting transcription initiation (using triptolide), we found that cluster formation is almost completely abrogated with the exception of HLBs as previously noted[10,40] (Fig. 3g, f, Movies S7, S10). In contrast, inhibition of elongation (using α-amanitin) leads to a moderate decrease in cluster density (Fig. 3g, f, Movies S6, S9) but a sharp increase in both the cluster lifetime and fraction of clusters in the long-lived population. In control embryos, the mean long-lived cluster lifetime is 6 ± 2 minutes in nc13 and 7 ± 1 minutes in nc14. However, in α-amanitin-injected embryos, the mean long-lived cluster lifetime increases to 9 ± 2 min in nc13 and 14 ± 3 minutes in nc14, reflecting a 1.5-fold increase in nc13 and a 2-fold increase in nc14 compared to the cluster lifetimes measured in control embryos (Fig. 3h, i, Figure S3. 2e). Furthermore, 67 ± 2% and 41 ± 2% of the clusters are long-lived in the α-amanitin-injected embryos in nc13 and nc14 respectively, reflecting a 1.4-fold and 1.2-fold increase over the corresponding fractions in control embryos (Fig. 3i). While these results show that transcriptional inhibition shifts the cluster lifetime distribution, we do not interpret short- and long-lived clusters as strictly corresponding to elongating or initiating RNAPII respectively. Rather, cluster lifetimes likely reflect a mixture of transcriptional states.

Together, these data show that transcription initiation is necessary for RNAPII cluster formation while elongation de-stabilizes clusters, perhaps due to a higher turnover of RNAPII at genes. Furthermore, clusters emerge well before the major wave of ZGA, and their properties change with an increase in transcriptional activity. While the absolute lifetimes of long-lived clusters stay relatively stable,

their fraction decreases markedly as transcriptional activity increases. To further investigate the regulation of clustering kinetics, we next quantified the molecular kinetics of RNAPII within clusters.

## RNAPII clusters are composed of elongating and kinetically confined molecules after ZGA

Consistent with our volumetric imaging data, our single molecule tracks exhibit clustering in vehicle-injected embryos which is reduced in α-amanitin injected embryos (Fig. 4a). We identified clusters based on the local density of trajectories and then filtered these clusters based on size and density to exclude likely HLBs and random accumulations (Figure. S4. 1a–f). By comparing RNAPII kinetics inside clusters versus the rest of the nucleoplasm (outside clusters), we found a 1.4 ± 0.4-fold enrichment of bound RNAPII trajectories inside clusters relative to outside clusters in both nc13 and nc14 in vehicle-injected embryos and a 2.5 ± 0.7-fold enrichment inside clusters relative to outside clusters in both nc13 and nc14 in α-amanitin injected embryos (Fig. 4b). To ensure that the observed enrichment inside clusters is not an artifact resulting from performing analysis within restricted regions of a nucleus, we quantified the bound fraction in control regions of the same size as the clusters but located randomly within the nucleus and not overlapping with clusters. In vehicle-injected embryos, the bound fraction inside clusters was 1.3 ± 0.3-fold higher than in control spots in both nc13 and nc14 (Fig. 4b). The bound fraction within clusters does not increase from nc13 to nc14 in α-amanitin injected embryos, similar to the global population (Fig. 2). In sharp contrast, in vehicle injected embryos, we found a 1.4 ± 0.2-fold increase in the bound population from nc13 to nc14. A decomposition of this bound population as done above (Fig. 2e) leads to a 6.1 ± 8.4-fold increase in the elongating population of RNAPII molecules within clusters from nc13 to nc14 (Fig. 4d).

To quantify how the kinetics of molecules in the non-chromatin bound (intermediate and free) populations are altered within clusters, we calculated the distribution of angles between three consecutive displacements and quantified changes in the anisotropy coefficient of tracks within and outside clusters[41] (Figure. S4. 1g–i). We found that the anisotropy coefficient inside clusters remained consistently higher than both outside clusters and control spots in nc13 and nc14 in both vehicle and α-amanitin injected embryos (Fig. 4c). The increased anisotropy in clusters suggests that molecules within them are exhibiting compact exploration kinetics and are likely kinetically confined through frequent rebinding or through an increase in local protein-protein interactions. Furthermore, in vehicle-injected embryos, the anisotropy coefficient inside clusters exhibits a ~ 1.3-fold increase from nc13 to nc14 (Fig. 4c). Overall, these analyses show there is an increase in the elongating fraction of RNAPII within clusters concurrent with an increase in the kinetic confinement of unbound RNAPII molecules during ZGA.

## A single RNAPII cluster persistently associates with a site of active transcription

To determine the behavior of RNAPII clusters at a specific site of active transcription, we examined clustering in the context of transcription using the MS2-MCP system. We visualized transcription using reporters for four different genes that are actively expressed in *Drosophila* embryos in nc13 and 14: eve (*eve*), hunchback (*hb*), snail (*sna*), and sog (*sog*). We performed two-color lattice light-sheet imaging, sequentially acquiring a volume in each channel every 9 seconds. For each of the four genes, we observed that a single RNAPII cluster was consistently associated with the *active* locus throughout a transcription burst (Fig. 5a, Figure. S5. 1a–c, Movie S11). This association was not visible when looking at a non-transcribing locus labelled using the ParS/ParB system (Figure. S5. 1d). The intensity of RNAPII in a 0.5 μm sphere around the active locus is highly correlated with the intensity of nascent transcripts in the same sphere during a transcriptional burst

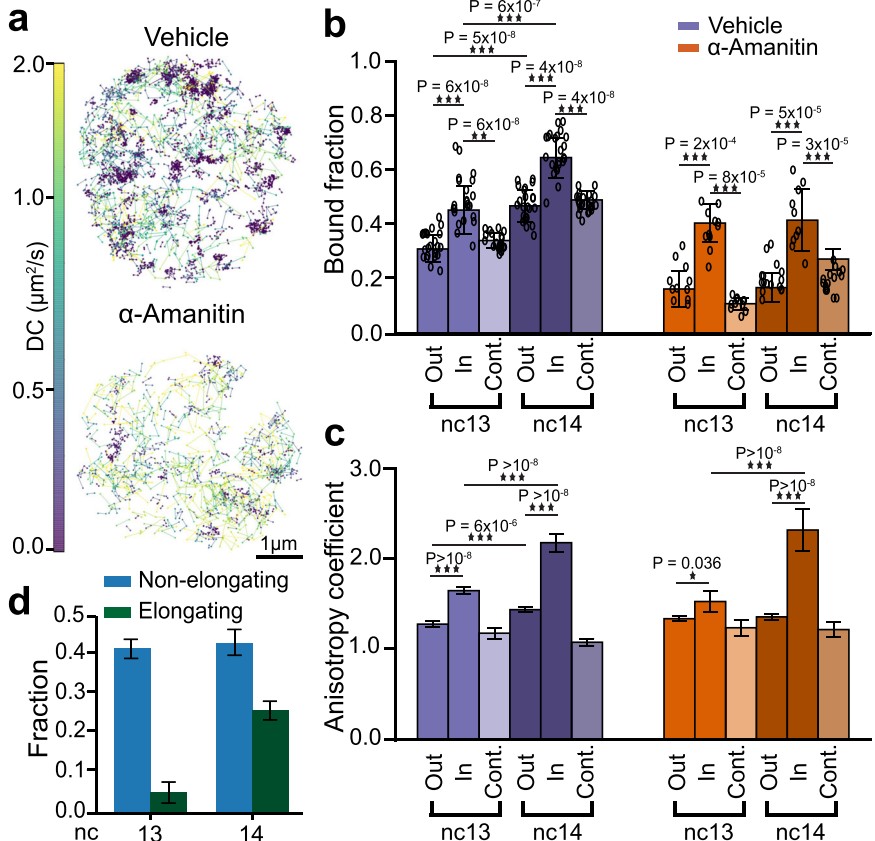

**Fig. 4 | Single molecule kinetics of RNAPII inside clusters during ZGA.**
**a** Representative nuclei showing single molecule tracks for mEos3.2-RPB1 color coded by the diffusion coefficient (colorbar to the left) for vehicle and α-amanitin injected embryos. Tracks are shown from a 60 sec total imaging window with an exposure time of 10 ms. **b** Bound fraction of RPB1 inside clusters, outside clusters and in control spots for vehicle and α-amanitin injected embryos in nc13 and nc14. Data points represent 23 individual fields of view for control embryos in both nc13 and nc14, and 11 and 12 individual fields of view for α-amanitin injected embryos in nc13 and nc14 respectively ( ~ 8–15 nuclei per field of view). Bars represent mean across the fields of view. Error bars show standard deviations.Two-sided Mann-Whitney U-test was performed to determine significance and following p-values were used: *$p < 0.05$, **$p < 0.01$, and ***$p < 0.001$. **c** Anisotropy coefficient of the trajectories outside clusters, inside clusters and inside control spots for the vehicle

and α-amanitin-injected embryos. Number of angles in vehicle-injected embryos are 13,107, 3070, and 5754 for outside clusters, inside clusters and in control spots respectively in nc13 and 10,540, 2057, and 6104 for the same categories in nc14. Number of angles in the α-amanitin-injected embryos are 2664, 995 and 2189 for outside clusters, inside clusters, and in control spots respectively in nc13 and 4807, 1125, and 2894 for the same categories in nc14. Bars represent the mean of the bootstrap analysis. Error bars represent standard deviation from bootstrapping analysis. Two-sided Mann-Whitney U-test was performed to determine significance and following p-values were used: *$p < 0.05$, **$p < 0.01$, and ***$p < 0.001$. **d** RNAPII bound fraction split into elongating and non-elongating (i.e., initiating + non-specifically binding) molecules in nc13 and 14. Bars reflect point estimates computed from the mean bound fractions. Error bars represent standard propagation of error. Source data are provided as a Source Data file.

(Fig. 5b, Fig. S5. 2b). Cross-correlation analysis of nascent transcripts and RNAPII intensity traces consistently exhibits a strong peak at zero lag times, indicating a consistent temporal synchronization (Fig. 5c, d, Figure. S5.2a). This peak was significantly reduced when calculating cross-correlations between the transcript traces and RNAPII traces from control spheres of the same volume but not overlapping with the locus, in each nucleus (Figure S5. 3). Thus, an RNAPII cluster accumulates at an active locus at the start of a transcription burst, and the concentration of RNAPII molecules within those clusters (as implied by the cluster intensity) is correlated with the level of transcriptional activity. As the RNAPII clusters become less intense, the transcriptional levels also drop, and eventually the cluster dissipates. Strikingly, we also observe an individual RNAPII cluster on each active sister chromatid post-replication when the alleles transiently separate allowing them to be resolved. This observation further indicates that a RNAPII cluster represents transcriptional activity at a single active gene (Fig. 5e).

If every active gene was regulated by an individual RNAPII cluster, we would expect to observe thousands of clusters per nucleus in nc14.

Since we only detect ~12 clusters per nucleus (Fig. 3), we reasoned that weaker clusters are not detectable above nuclear background with our imaging methods. To test this hypothesis, we developed computer simulations that model the association of RNAPII molecules at an actively transcribing gene locus, as it would appear under our microscope (see Methods). The simulations generated both microscope-like "images" and RNAPII intensity traces during transcriptional bursts, closely resembling our reporter gene data. By varying the probability of RNAPII association with an active gene ($k_{on}$) to mimic enhancers of different strengths, the simulations produced a broad range of burst parameters, consistent with our experimental measurements at our reporter genes (Fig. 5f–h). Clusters formed at the gene in the simulations are qualitatively similar to those seen at our reporter genes. Simulated clusters at loci with lower $k_{on}$ exhibit slower loading rate and shorter burst duration, leading to the clusters being indistinguishable from the nuclear background (Fig. 5i, Movie S12). While cluster detectability can also be influenced by the promoter on-time ($t_{on}$) and gene length (Movie S13), since these parameters are constant for our reporter gene constructs, they were not varied in our simulations.

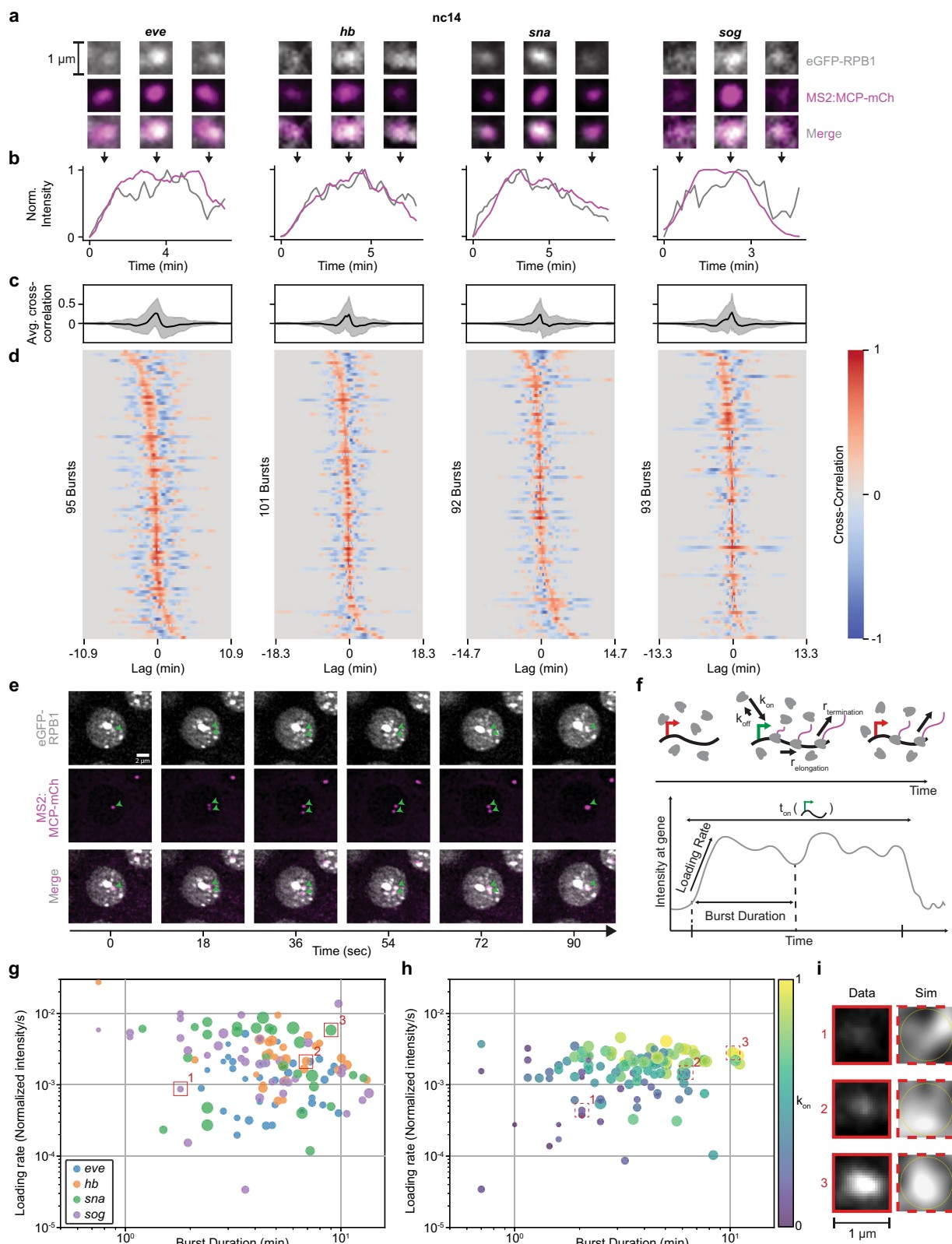

Overall, our simulations show that varying $k_{on}$ is sufficient to recapitulate the spread in our experimental burst data.

## Discussion

Here we show that during the widespread increase in transcriptional activity during ZGA, RNAPII clusters transform from collections of largely initiating molecules, to primarily elongating molecules. This

framework helps reconcile contradictory reports that describe RNAPII clusters as either initiation-specific[11,12,42] or only elongation-associated[13]. Our data instead suggest that RNAPII clusters prime transcriptional activity prior to ZGA, and largely represent regions of high transcriptional activity after ZGA (Fig. 6). Furthermore, perturbations to the transcription cycle and live imaging of transcription show that RNAPII cluster formation is dependent on pre-elongation

**Fig. 5 | RNAPII clusters "burst" concomitantly with a transcription burst.**
**a** Representative images of eGFP-RPB1 (grey) with MCP-mCh (pink) labelling active transcription at four reporter genes in nc14 at the beginning, middle and end of a transcription burst. **b** Normalized intensity of eGFP-RPB1 (grey) and MCP-mCh (pink) intensity in a 1.2 μm diameter circle around the MS2 spot at each of the four genes. The images in (**a**) are from the same burst shown. **c** Average cross-correlation of eGFP-RPB1 and MCP-mCh intensities around the MS2 spot at each of the four genes in nuclei in nc14, from three biological replicates. Gray shaded area in line plots represent standard deviation. RPB1 intensity was normalized to reflect enrichment above nuclear background. **d** Individual cross-correlations of eGFP-RPB1 and MCP-mCh intensities around the MS2 spot at each of the four genes in nuclei in nc14, from three biological replicates. **e** Representative images of eGFP-RPB1 (grey) with MCP-mCh (pink) labelling active transcription of the same reporter on sister chromatids. **f** Schematic showing RNAPII molecules transcribing a gene (top) and the resultant intensity trace of RNAPII at the gene (bottom). **g** Loading rate plotted against burst duration for each RNAPII burst. The marker size reflects the maximum RNAPII intensity during the burst. **h** Loading rate plotted against burst duration for each RNAPII burst from simulated data where $k_{on}$ is varied. The marker size reflects the maximum RNAPII intensity during the burst. **i** Comparison of RNAPII clusters from experimental and simulated data. Source data are provided as a Source Data file.

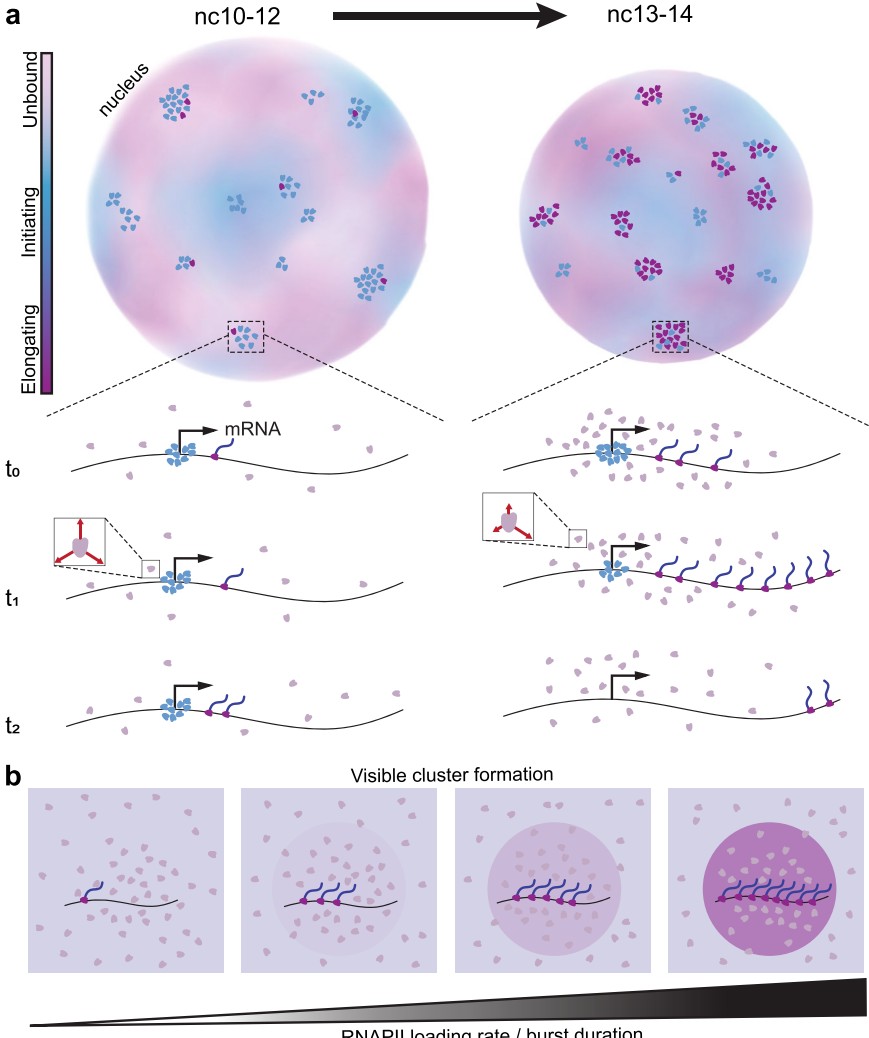

**Fig. 6 | Functional transformation of RNAPII clusters during ZGA. a** In earlier nuclear cycles (nc10-12), RNAPII clusters are primarily comprised of initiating RNAPII molecules. As transcription increases in later cycles (nc13-14) clusters are associated with elongation. When transcription activity peaks in nc14 molecules within clusters exhibit an increased kinetic confinement as indicated by red arrows in the inset. **b** The RNAPII loading rate dictates in conjunction with the burst duration determines whether a cluster of molecules is visible over the background.

steps, yet once polymerases within clusters are released into elongation, they persist at an active locus during the entire duration of a transcription burst (Fig. 6).

In the nuclear cycles prior to the major wave of ZGA (nc10-12), RNAPII clusters exhibit lifetimes comparable to interphase durations (Fig. 3d), consistent with low turnover of molecules and a high propensity for aborted transcription[19]. This early-stage clustering may poise developmental genes for activation, as has been described during dorso-ventral patterning at ZGA[43] and later during tissue specification[44]. Beginning in nc13, the fraction of long-lived clusters declines sharply, while the nuclear cluster density increases (Fig. 3c). In nc14, we also observe an increase in the anisotropy of non-bound RNAPII molecules within clusters, which suggests a kinetic confinement of molecules near active transcription sites (Fig. 4). This confinement becomes more pronounced upon transcription inhibition with α-amanitin, suggesting that the increased recycling of RNAPII molecules at promoters can extend cluster lifetimes without altering the overall bound fraction[30,34]. Though less likely given our data, this

confinement could also reflect the role of protein-protein interactions in mediating RNAPII clustering through its C-terminal domain[45–47]. Together, these changes indicate a transition toward elongation-dominated clusters. The increase in elongation within clusters is supported by an increase in Ser2 phosphorylation signal in RNAPII clusters (marking elongating RNAPII molecules; Figure S3. 2) in nc14 compared to early cycles and by the reversion of lifetimes and densities upon elongation inhibition (Fig. 3f–h). Fixed-cell studies in mammalian embryos have also shown Ser2 phosphorylated molecules getting progressively enriched along the gene body while the presence of only Ser5 phosphorylated molecules decreases during zygotic genome activation[48]. Furthermore, in *Drosophila* embryos, RNAPII clusters during nuclear cycle 12 (well before ZGA) are strongly enriched for Ser5 phosphorylation but show minimal Ser2 phosphorylation signal, consistent with a primarily initiating state[49].

Similar increases in RNAPII cluster lifetimes following elongation inhibition via DRB or flavopiridol have been observed in cell culture studies[12]. Additionally, the dissolution of transcriptional condensates in response to increased transcriptional activity has been proposed to arise from changes in electrostatic forces from local RNA concentrations[18]. The high correlation of local RNAPII intensity and transcriptional output we measure (Fig. 5) suggests that the shift toward shorter-lived clusters may simply reflect an increase in the number of RNAPII molecules being released into the gene body and then completing transcription, rather than higher-order regulatory mechanisms. In this context, our use of the term "cluster" does not imply a specific assembly mechanism such as phase separation. Indeed, our simulations (Fig. 5) show that bursts of RNAPII loading and elongation alone can generate RNAPII enrichment at an active gene without invoking a biomolecular condensate model. This is consistent with recent findings on the transcription factor GAF, where foci that visually resemble condensates were shown to arise purely from high chromatin occupancy at satellite repeat clusters[50]. Furthermore, our observed intensity correlation aligns with previous models of transcriptional bursting in which the burst shape represents the kinetics of RNAPII loading and elongation at a gene body[51,52].

The strong correlation between mRNA production at a single active gene and RNAPII cluster intensity suggests that each cluster could be associated with a single gene. Yet, despite the increase in transcriptional activity in nc14, the nuclear cluster density does not increase compared to nc13 (Fig. 3c), and the number of countable clusters per nucleus is far lower than the number of transcribed genes. This discrepancy could reflect two non-mutually exclusive possibilities: (1) most genes fail to recruit sufficient polymerases at a time to form visible clusters above the nuclear background in live imaging, or (2) clusters occasionally contain multiple co-bursting genes[53]. Consistent with the first possibility, our simulations (Fig. 5f–i) show that the number of RNAPII molecules engaged with the active locus is enough to mediate the detectability of an RNAPII cluster over the background. For endogenous genes, this number can be influenced not only by a variable $k_{on}$, but also by the length and on-time of the gene (Movie S13). Furthermore, when we imaged sister chromatids post replication each containing an MS2 reporter allele, we observed distinct RNAPII clusters at each locus rather than a shared RNAPII cluster (Fig. 5e). Still, we cannot completely exclude multiple genes within an RNAPII cluster as also supported by Hi-C studies in *Drosophila* embryos showing that RNAPII is enriched at the boundaries of Topologically Associated Domains (TAD) and mediates inter-TAD compaction during development[32].

Overall, our observations align well with an individualistic model of gene regulation by clusters[10], with a single RNAPII cluster per gene. This stable association contrasts with reports of larger RNAPII condensates engaging in transient "kiss-and-run" interactions with genes to enhance transcriptional bursting[16]. These larger condensates are potentially equivalent to histone locus bodies (HLBs), as implicated by recent proteomics profiling[54]. Consistent with this, HLBs in our own and others[10] experiments persist even after inhibition of elongation, and may influence transcription differently than the majority of clusters that we focus on in this work. Our findings also contrast with previous observations in *Drosophila* embryos that reported RNAPII cluster intensities peak before transcription and decline well before burst termination, implying that only a minority of recruited polymerases enter into productive elongation[17]. This discrepancy may arise from methodological differences, as in this previous study[17] the aggregate intensity of all RNAPII clusters in the nucleus was analyzed, whereas we directly measured the intensity of RNAPII clusters at an actively transcribing gene. Additionally, the study focused on nc12 where both transcription levels and the fraction of elongating polymerases is lower compared to nc13 and 14.

Taken together, our data suggest that RNAPII clusters are not limited to a single functional state but are functionally labile. In earlier developmental cycles, they are composed primarily of initiating molecules, transitioning in later cycles to primarily being elongating molecules. Furthermore, our results highlight that RNAPII clusters are stably associated with transcription sites. Future studies should explore the molecular triggers for these functional transitions, the initial drivers of RNAPII cluster formation in relation to other core components of the pre-initiation complex, and the role of changes in local chromatin density and topology in modulating cluster lifetimes.

## Methods
### Western Blots
Embryos were collected 30 min after egg laying and incubated for 75 min at 25 °C. They were dechorionated with 50% bleach and staged in Halocarbon Oil 27. Approximately 100–125 embryos were harvested in 1× PBS supplemented with protease inhibitor cocktail (PIC) and homogenized on ice in a 4× Laemmli SDS sample buffer containing 5% β-mercaptoethanol using disposable pestles. Lysates were heated to 95 °C for 5 min, cooled to room temperature, and centrifuged at 16,000 rpm (31,483 x g) for 10 min at 4 °C. The supernatant was transferred to new 1.5 mL tubes and stored at −20 °C until use. Protein samples were divided into two equal volumes and separated on polyacrylamide gels according to target size. For Ser5 blots, proteins were resolved on an 8% acrylamide gel at 150 V for 5 h. For RPB1-CTD blots, proteins were resolved on a 6% acrylamide gel to improve separation of the high–molecular weight protein, and gels used for expression analyses were run overnight (18 h) at 40 V in an ice bath in a 4 °C cold room. For β-tubulin and α-tubulin controls, proteins were separated on an 8% acrylamide gel for 1 hour under the same conditions as the Ser5 blots. Proteins were transferred overnight at 20 mA at 4 °C. A Tris-glycine transfer buffer containing 10% methanol and 0.05% SDS was used for RPB1-CTD blots, whereas a buffer containing 20% methanol without SDS was used for Ser5 and tubulin blots. Nitrocellulose membranes were employed for Ser5 and tubulin, while PVDF membranes were used for RPB1-CTD to improve retention and detection of the large CTD protein. All membranes were blocked for 30 min with 3% milk in 1× TBST (1× TBS, 0.2% Tween) at room temperature. Membranes were incubated overnight at 4 °C with primary antibodies: rabbit anti-Ser5 (Abcam, ab5131; 1:250), mouse anti-RPB1 CTD (Cell Signaling Technologies, 2629 T, 1:1000), rabbit anti-α-tubulin (Cell Signaling Technologies, 2144, 1:1000), or rabbit anti–β-tubulin (Abcam, ab179513; 1:1000). After washing with 1× TBST, blots were incubated with HRP-conjugated anti-rabbit or anti-mouse secondary antibodies (1:10,000) for 1 h at room temperature. Chemiluminescent detection was performed using ECL substrate, and signals were visualized with a BioRad ChemiDoc imaging system. Band intensities were quantified using ImageLab volume measurement software.

## Automated Western Blots Using Jess

Embryos were collected after 30 minutes of egg laying and incubated for 75 minutes at 25 °C. Embryos were then dechorionated with 50% bleach and staged in Halocarbon Oil 27. Approximately 250 embryos of each genotype were collected in 1.5 mL microcentrifuge tubes and flash-frozen on dry ice. Embryos were washed twice, resuspended in 500 μL of M-PER™ lysis buffer supplemented with protease inhibitor cocktail (PIC), and centrifuged at ~300 × g to pellet the embryos. Subsequently, 475 μL of lysis buffer was removed, and embryos were homogenized on ice using pre-chilled disposable pestles. Homogenates were centrifuged at ~20,000 × g for 5 min at 4 °C. Pellets were discarded, and supernatants were transferred to fresh microcentrifuge tubes. Total protein concentration for each sample was quantified using a Bradford assay. Protein samples were diluted to a final concentration of 0.5 mg mL$^{-1}$ using 0.1× Jess Sample Buffer.

Due to the large molecular weight difference between RPB1 and the loading control, samples were analyzed using two separate Simple Western™ Jess separation modules. RPB1 separation was performed using the 66–440 kDa Separation Module under standard Jess default assay conditions. Proteins were immunodetected using mouse anti-RPB1 CTD Pan primary antibody (1:10 dilution; Cell Signaling Technology, 2629 T), followed by HRP-conjugated anti-mouse secondary antibody and chemiluminescent detection using the Jess HRP Detection Reagent. For the loading control, equivalent amounts of protein from each genotype were analyzed using the 12–230 kDa Separation Module and detected with anti-α-tubulin primary antibody (1:10 dilution; Cell Signaling Technology, 2144), followed by HRP-conjugated secondary antibody and Jess HRP Detection Reagent.

## Immunofluorescence Imaging

Embryo collection and staging were performed as described in the western blot protocol. Dechorionated embryos were transferred into fixation solution (4% PFA in 1X PBS) using a paintbrush, and an equal volume of heptane was added. The embryos were vortexed at low speed for 30 min. To remove the vitelline membrane, 500 μl of methanol was added, and the embryos were vortexed for 30 s. All methanol was removed and replaced with fresh methanol. The embryos were then washed in 0.1% PBST four times for 5–10 min each, followed by blocking in 1 ml of fresh 5% NHS for 30 min at room temperature. The embryos were evenly split into two tubes; each tube was incubated with a 1:100 dilution of the primary antibody (Ser5 or Ser2) overnight on a nutator at 4 °C. After incubation, the embryos were washed four times for 5–10 minutes each with 0.1% PBST, then incubated with a 1:500 dilution of the secondary antibody for 1 hour at room temperature. The secondary antibody was rinsed off, and the embryos were washed four more times before mounting on glass slides with VectaShield. Images were acquired using a ZEISS LSM 900 confocal microscope. A LD LCI Plan-Apochromat 40x/1.2 Imm Korr DIC M27 objective was used to acquire single-plane Fast Airyscan super-resolution images of a 40 μm x 40 μm area (pixel size 49.7 nm; 4x digital zoom, SR-2Y sampling mode). The sequential frame scanning mode with a pixel dwell time of 42 μs was used.

## EUTP microinjection and Click-iT reaction

Embryos were collected, dechorionated, mounted, and then desiccated for 7 minutes in a desiccation chamber before covering them with a thin layer of Halocarbon oil and staging them. Embryos were injected with 1:10 dilution EUTP in NF water or 1:1 of diluted EUTP to 0.5 mg/ml of Triptolide dissolved in DMSO 15–20 min post division into nc13. Injected embryos were carefully washed and transferred to a microcentrifuge tube containing 500 μl each of heptane and 37% formaldehyde as described previously[17] using a paintbrush dipped in heptane. The tube was shaken for 1 minute and left standing for 10 minutes, following which the vitelline membrane was removed as described above and stored in methanol. The embryos were then

washed in 0.1% PBST four times for 5–10 min each and incubated on a nutator for 30 min with the Click-iT reaction cocktail as described in the manufacturer's protocol (Invitrogen). The embryos were then washed with the provided wash buffer and PBST three times for 5–10 min each and then mounted on glass slides with VectaShield. Images were acquired using a ZEISS LSM 880 confocal microscope. A Plan-Apochromat 63x/1.4 Oil DIC M27 objective was used to acquire z-stacks in a 45 μm x 45 μm area (pixel size 56.8 nm) with an interval of 300 nm between slices. The sequential bidirectional line scanning mode was used with a pixel dwell time of 2.65 μs with line averaging done over 4 scans.

## Embryo collection and mounting for live-imaging

Embryos were incubated at 25 °C for 45 min and transferred from an apple juice-agar collection plate to a cell strainer immersed in a petri dish of water. Embryos were dechorionated by incubating in 6% sodium hypochlorite for 51 s, then immediately rinsed with distilled water. Embryos were then transferred and positioned in rows on an agar pad using a fine paintbrush and dissecting needle under a dissection microscope. A 25 mm glass coverslip was coated with 8 μl of double-sided Scotch tape dissolved in heptane to make it adhesive. Once the heptane was fully evaporated, embryos were transferred to the coverslip by gently tapping. A drop of distilled water was added to hydrate the embryos on the coverslip.

## Embryo microinjection

Embryos were collected, mounted, and imaged with a lattice light-sheet microscope to determine their developmental stage. Embryos in nc12 were identified for injection and the sample was transferred for injection. Distilled water was removed from the coverslip using a Kimwipe, and a drop of Halocarbon oil 27 (Sigma Aldrich) was added to the embryos. Microinjections were conducted using an Eppendorf Femtojet mounted on a Leica DMi8 Microscope. A Sutter P-87 needle puller was programmed to pull "medium taper" needles at Heat = 915 °C, Pull = 80, Velocity = 28 and Time = 250 using thin-wall borosilicate capillaries (WPI TW100F-4). Pulled needles were calibrated for injection by ensuring they consecutively produced 10 bubbles of similar diameters in halocarbon oil. Pulled microinjection needles were loaded with 3.5 μl of either α-amanitin, triptolide or the vehicle control to be injected. Step injection was conducted at 0.1 sec, with a compensatory pressure of 15 kPa. Post injection, halocarbon oil was gently washed off the embryos with distilled water and remounted on the lattice light-sheet microscope for imaging of the injected embryos. α-amanitin (Sigma Aldrich) was dissolved in water at a concentration of 0.5 mg/ml. Triptolide (MedChem Express) was dissolved in DMSO at 1 mg/ml and diluted with water at a 1:1 ratio. DMSO was diluted in water in a 1:1 ratio for the negative control experiment. Each embryo was injected with approximately 0.2 nl of the solution[32]. All embryos were mounted for imaging within five minutes of microinjection.

## Microinjection Survival Assay

Mounted H2B-eGFP embryos were imaged on a Leica DMi8 Microscope and injected in different nuclear cycles to determine survivability. Nuclear cycles were determined by bulk histone H2B-eGFP markers in the nuclei. Microinjections were conducted and then development to gastrulation was noted. Based on the survival assay, it was determined that embryos that were injected with drugs in nc12 or later showed normal development until the mid-point of nc14 as evidenced by regular division cycles and normal morphology. Embryos injected with drugs prior to nc12 showed significant developmental defects within ~10 min, well before nc14. Embryos that were injected with the vehicle did not show any defects and proceeded to gastrulation.

## Light-sheet microscope optical paths and configuration

The lattice light-sheet microscope[55] used in this work is a modified, home-built implementation based on the adaptive optics-equipped lattice light-sheet system developed by the Betzig lab at HHMI Janelia Research Campus and UC Berkeley[56]. For experiments in this work the following laser lines were used: 405 nm (Coherent), 488 nm (MPB Communications Inc.) and 589 nm (MPB Communications Inc.). Briefly: the laser lines were expanded to a diameter of 2 mm and combined using a series of dichroic mirrors, passed through a Half-wave plate (Bolder Vision Optik) to adjust polarization and relayed into an acousto-optic tunable filter (Quanta-Tech, AA Opto Electronic) to select wavelength and modulate power. The output from the AOTF was then either sent to an optical path to generate a lattice light-sheet excitation pattern or a multi-gaussian beam excitation pattern. For lattice light-sheet generation the collimated laser beams were expanded along a single dimension using a Powell Lens (Laserline Optics Canada) and then the width of the expanded beam was adjusted and re-collimated using a pair of cylindrical lenses (25 mm diameter; Thorlabs). This stripe of collimated light was relayed onto a grayscale Spatial Light Modulator (SLM; Meadowlark Optics, AVR Optics) after passing through a second Half-wave plate. The SLM is conjugate to the sample-plane of the microscope. The diffracted light from the SLM was passed through a lens to project its Fourier transform onto a custom built annular mask to select the minimum and maximum numerical aperture of the light-sheet and block unwanted diffraction orders from the SLM. The annular mask is positioned in the pupil plane of the excitation light path. The annular mask plane was demagnified and projected onto a resonant galvanometer (Cambridge Technology, Novanta Photonics) conjugate to the sample plane. The resonant galvanometer was used to mitigate shadowing artifacts and other inhomogeneities in the light-sheet by introducing a slight wobble in the excitation angle. The light was then projected onto a pair of galvanometer scanning mirrors (Cambridge Technology, Novanta Photonics) (conjugated to the pupil plane) for scanning the light-sheet along x and z optical axes in the excitation coordinate plane. Finally, an excitation objective (Thorlabs, TL20X-MPL) was used to focus the light-sheet onto the sample. The emitted fluorescence was collected by a detection objective oriented orthogonally to the excitation objective (Zeiss, 20×, 1.0 NA), and projected onto a Deformable mirror (ALPAO) positioned in a pupil of the detection path. The light was then split using a dichroic beam splitter (Semrock Di03-R561-t3-25×36) and imaged onto two sCMOS detectors (Hamamatsu ORCA Fusion). The first camera had a green emission filter (Semrock FF03-525/50-25) and a notch filter (Chroma ZET488NF) to reject laser light, and the second camera had a red emission filter (Semrock FF01-593/46-25) and a notch filter (Chroma ZET561NF) to reject laser light. Optical aberrations in the detection path were corrected by adjusting the deformable mirror (Alpao), as previously described in ref. [57].

## Single-molecule imaging and tracking using light-sheet microscopy

For single-molecule imaging a Gaussian light sheet was used (see the section on "Light-sheet microscope optical paths and configuration" for more details). Briefly, after passing through the AOTF, the beam was expanded using a Powell lens following which it was relayed onto a custom mask for filtering. The filtered sheet was then projected onto a resonant galvanometer (Cambridge Technology, Novanta Photonics). Finally, an excitation objective (Thorlabs, TL20X-MPL) is used to focus the light-sheet onto the sample. The detection path for the emitted light is the same as described above for the lattice light-sheet configuration. For this light path, the SLM was bypassed to maximize laser power at the excitation objective.

For all the single-molecule imaging, the 405 nm laser line was kept on constantly during the acquisition period for photo-switching and the 561 nm laser line was used for excitation. Data

was acquired at 10 msec exposure time. Excitation laser power was empirically optimized to maximize contrast for single-molecule tracking while minimizing photobleaching. The power of the photoswitching laser was also optimized to maintain a low enough detection density for tracking. The excitation laser power used was 13 mW and switching laser power used was 1 µW as measured at the back focal plane of the excitation objective. A total of 8,000 frames were acquired corresponding to 80 s of total imaging time. The acquisition length was optimized such that multiple fields of views could be imaged within each short interphase time while also capturing a sufficient number of trajectories at each position. To optimally position the embryo in the light sheet and to keep track of cell-cycle phase, and nuclear cycle, H2B-eGFP was used.

## Volumetric imaging using lattice light-sheet microscopy

Live volumetric imaging of the eGFP-RPB1 line was done by using a multi-bessel lattice sheet with maximum numerical aperture to minimum numerical aperture ratio of 0.4/0.3. A 488 nm laser was used for the volumetric imaging with laser power of 0.512 µW and exposure time of 60 msec. For each time point, 64 slices were imaged at a thickness of 300 nm per slice for a total depth of 18.9 µm. Each volume was acquired every 10 sec. For sequential two-color imaging of the eGFP-RPB1 with genes tagged with MS2:MCP-mCherry or ParS:ParB-mCherry, the 488 nm laser was used for exciting eGFP while the 589 nm was used for exciting mCherry. Laser powers of 0.512 µW and 1.40 µW were used for the 488 nm and 589 nm wavelengths respectively at an exposure time of 40 msec. The two channels were acquired sequentially for a total volume of 18.9 µm. For each channel, 64 slices were imaged at a thickness of 300 nm per slice. Each volume was acquired every 9 s.

## Processing of single-molecule tracking data for quantifying diffusion kinetics

To process the SMT data, we first removed the initial ~2,000 frames affected by fluorophore bleaching. Maximum projections were generated to obtain nuclear masks. Localization and tracking of the single molecule trajectories was performed using the open source software package quot (https://github.com/alecheckert/quot). For the 10 msec data the following key parameters on quot were used: (i) for detection - method = "identity", Spot detection settings = "llr", k = 1.3, w = 15, t = 21, (ii) for localization - method = "ls_int_gaussian", window size = 9, sigma = 1.0, ridge = 0.0001, maximum iterations = 10, damp = 0.3, camera gain = 1, camera background = 108 (iii) for tracking - method = "euclidean", pixel size (in µm) = 0.108, search radius = 0.7 µm, maximum blinking = 0 frames. Once we obtained our list of tracked molecules from quot, we filtered them using the nuclear mask as described earlier to ensure we analyzed only trajectories that were occurring inside the nucleus.

## Analysis of single-molecule tracking data

To analyze the 10 msec single molecule trajectories to infer the diffusion kinetics, we used a variational Bayesian method called State Array based Single Particle Tracking[27]. This method does not a priori assume a specific number of diffusive states. Instead, it processes all the recorded trajectories and selects a model that comprehensively describes the trajectories with the minimum combination of state parameters (diffusion coefficient and localization error) as possible. Consequently, it produces average posterior state occupancies for a state array evaluated on the experimental trajectories. For the model selection, we set a range of 100 diffusion coefficients extending from 0.001 µm²/s to 100 µm²/s representing a range of biologically plausible diffusion coefficients. This package is publicly available at https://saspt.readthedocs.io/en/latest/. For our analysis, we used the following specific parameters in the saSPT program:

Likelihood type: RBME Pixel size: 0.108 (1 pixel = 0.108 μm) Frame interval: 0.0125433 msec Concentration parameters: 1 Max iterations: 200 Split Size: maximum trajectory length Sample Size: number of trajectories

Using saSPT, we were able to generate the diffusion coefficient occupancy plots for each protein category. We identified the local maxima and minima by taking the derivative of the occupancy plots and assigned the different kinetic states as follows: (i) the first state extends from the lowest DC value of 0.001 μm²/s to the first local minima detected; (ii) the last state extends from the last local minima detected to the highest DC value of 100 μm²/s; (iii) all other states extend between consecutive minima. We used the minima obtained from the vehicle injection in each nuclear cycle to assign the states for the drug conditions in that corresponding nuclear cycle.

Furthermore, in order to assign individual diffusion coefficients to each trajectory for downstream analysis, we first weighted the range of diffusion coefficients by their corresponding occupancies for each trajectory. We calculated the geometric mean of these weighted coefficients which was then assigned to each trajectory. To estimate the localization error in our single-molecule tracking data, we analyzed the subset of H2B-mEos3.2 trajectories with the lowest diffusion coefficients. We calculated the mean of their consecutive displacements over time and determined that the localization error was 30 nm.

To estimate the relative contributions of elongating and non-elongating RNAPII molecules within the total bound fraction, we used a subtraction-based decomposition approach. For each nuclear cycle, the bound fraction measured in triptolide-injected embryos was assigned to the non-specific component. The difference between the bound fractions in α-amanitin and triptolide-injected embryos was attributed to the initiating component. The elongating fraction was thus calculated as the remainder, by subtracting the α-amanitin bound fraction from the total bound fraction in the vehicle-injected embryos. This analysis was performed separately for nc13 and nc14 to obtain the fraction change in the elongating population. Standard error propagation was used to estimate the error bars for each component.

For the anisotropy analysis, we first filtered the data sets to remove all the bound trajectories since they can bias the angle distribution. We then calculated the angle between all consecutive jumps (Figure S4. 1h,i) while excluding all jumps that were less than 0.2 μm based on the jump distribution of H2B (Figure S4. 1g). We then calculated the fold-anisotropy metric as the probability of observing a backward jump (with an angle between jumps in the range [180° - 30°,180° + 30°]), divided by the probability of observing a forward jump (with an angle between jumps in the range [0−30,0 + 30]).

To determine the clustering behavior from the single molecule trajectories, the average position of the trajectories were first calculated by considering the mean x and y locations. Average trajectory positions were considered because we wanted to quantify the diffusion kinetics inside the clusters and hence needed to assign diffusion coefficients to individual trajectories. We then used density-based spatial clustering of applications with noise (DBSCAN) to determine the location of the clusters and the trajectories that comprised the cluster. For the DBSCAN analysis, we specified that there had to be a minimum of 15 points ("min. samples") to define a cluster and the maximum distance between any two points in the cluster was 0.2 μm ("max. distance"). These values were selected by initially plotting the number of clusters as a function of the max. distance to determine where the plot showed an elbow, i.e. after what threshold value of max distance did the number of clusters did not change significantly, a common analysis technique used in DBSCAN to determine the appropriate parameters (Figure S4. 1b). The parameters chosen were further validated by qualitatively comparing the DBSCAN predictions to the raw spatial maps of the single molecule trajectories. These values were kept constant between all the protein categories and nuclear cycles.

For any nucleus in which clusters were identified by DBSCAN, we also defined thirty control spots per cluster identified. These control spots were assigned such that they satisfied the following four criteria: (i) they did not overlap with any of the DBSCAN identified clusters in that nucleus, (ii) they did not overlap with each other and (iii) they were of the same size as the DBSCAN identified clusters and (iv) they were within the nuclear mask. The kinetics inside these assigned spots were used as a negative control against the measurement of bound fraction and kinetics inside the DBSCAN identified clusters. Furthermore, the timescale of anisotropy measurements (~20–150 msec from individual trajectories) is much shorter than the timescale of bulk cluster motion which occurs over tens of seconds.

Single molecule trajectories were taken from a minimum of 3 independent embryos for all the conditions and nuclear cycles analyzed. Detailed trajectory number and the number of fields of views from which the trajectories were compiled can be found in each figure caption. The diffusion coefficient occupancy plots for each condition and cycle was obtained after concatenating the data sets from all the movies acquired. The standard deviation of the bound fractions was calculated across the individual fields of views. The standard deviation for the fold-anisotropy was calculated by bootstrapping 50% of the data 20 individual times.

## Compositional data analysis (CoDA) of SMT fractions

In order to compare the relative proportions of bound, intermediate, and fast RNAPII fractions across different conditions, we applied a compositional data analysis framework. For each field of view we computed the geometric mean of the three fractional values and then applied a centered log-ratio (CLR) transformation to each component. The CLR-transformed values were then used for statistical testing. Pairwise comparisons between conditions were performed using the Mann-Whitney U test on the transformed values. This approach accounts for the compositional constraint that the three fractions must sum to one, which violates the independence assumption required for standard non-transformed statistical tests. For the kinetics inside the clusters (Fig. 4), only the bound fractions are compared independently, and thus a standard statistical test was applied without CLR transformation.

## Analysis of volumetric imaging data to quantify cluster properties

Volumetric imaging data were pre-processed using GPU-accelerated 3D image deconvolution with CUDA[58] before downstream analysis. All datasets were deconvolved using input PSF taken by bead images on the LLSM and using ten iterations for Richardson-Lucy based deconvolution. We used a custom Imaris converter leveraging fast [Tiff or Zarr] file readers to generate Imaris files for data visualization and rendering[59]. All volumetric images shown in the figures of this manuscript are deconvolved images. After deconvolution, images were subjected to nuclear segmentation. To segment nuclei in our dataset, we created a custom model using Cellpose 2.0[60]. Ground truth data was generated using a mixture of micro-Sam[61], a Napari plugin for segment anything[61] and manual correction on wildtype eGFP-RPB1 images. This dataset was then used to train Cellpose 2.0. The resulting model was then used to segment all slices of each acquired dataset individually. A custom post-processing pipeline was then utilized in order to stitch the individual slices back together and to interpolate any slices of called objects that were missed in segmentation. We then implemented a nearest neighbor algorithm to track nuclei over the course of interphase in each nuclear cycle.

To quantify cluster properties, we modified a custom analysis pipeline we previously used[26]. In this pipeline, nuclei were first normalized to their mean intensity to assess local enrichments of TFs above nuclear background. This pipeline segments clusters by using a median filter to remove noise followed by image erosion and

reconstruction. The reconstructed image was subtracted from the median filtered image to first create a binary mask of high-density regions. Then, we called local maxima peaks to be used as markers for watershed segmentation in order to separate clusters that might be fused together in the binary mask. We then used region props to quantify different properties of the clusters such as integrated intensity, mean intensity, and size. This pipeline indiscriminately segments out all clusters including the larger histone locus bodies (HLBs).

To distinguish smaller clusters from the larger clusters surrounding HLBs, we segmented HLB clusters by using a difference of Gaussians (sigma high = 5, sigma low = 1) followed by a percentile threshold set by visual inspection (99.95%). Knowing that there are only two or less HLBs in a single nucleus in any given frame, we identified and removed the segmented clusters that overlapped with the segmented HLBs and had the highest measured cluster enrichment. Thereby, resulting in separate masks for eGFP-RPB1 clusters and HLBs. We then filtered these clusters (without HLBs) by their mean enrichment. Using empirical testing, we found that an enrichment of about 1.65 was appropriate for segmenting the RPB1 clusters without picking up on background noise. We used this cut off as a means to clear any cluster that might have a minimal and non-significant enrichment compared to the nuclear mean intensity. To calculate cluster density we divided the total number of clusters in each nucleus by the corresponding nuclear volume.

### Analysis of MS2-RNAPII interactions

The center coordinates of each MS2 spot was identified by first segmenting each spot. A difference of gaussians (sigmas = 1.5 and 6) followed by a percentile threshold (each threshold manually verified by visualizing multiple frames throughout the time course) was used. Small objects below 6 pixels were removed as noise. Cytoplasmic objects were removed if they were outside the nuclear labels. To account for multiple MS2 spots per nucleus (due to replication), the highest intensity spot was selected for tracking across frames. From this segmentation, the MS2 spot center was then calculated by using the segmented object's weighted centroid position. These coordinates were used to generate tiff stacks that were a 3-slice max-projection in z around the z-slice with the brightest MS2 spot at each time point. The coordinates were manually edited to correct any errors in spot tracking. RNAPII and MCP intensity traces were calculated as the integrated intensity in a 11 pixel (1.2 μm) diameter circle around the MS2 spot at each time point. The MS2 spot was taken as a 2 × 2 pixel square with the highest MCP intensity in each max-projected z-slice. The traces were smoothed and the RNAPII traces were normalized to reflect enrichment over the nuclear background. Individual bursts were segmented from the RNAPII traces using a local minima detection on the MCP trace, and burst durations and loading rates were calculated for all bursts longer than 30 s. Full, discrete, linear cross-correlations were computed in Python by centering the intensity traces and using the 'correlate' function from the scipy.signal package, with the 'full' mode and 'direct' method specified as arguments. These were normalized by dividing by the product of the signal standard deviations. The value of these direct correlation at lag = 0 is equivalent to the Pearson correlation coefficient.

Kymographs were generated from these tiff stacks in ImageJ. A 1.2 × 4.4 μm (11 × 41 pixel) slice around the MS2 spot was selected and a kymograph was made for each row. The 11 resulting kymographs were then max-projected to create the final max-projected kymograph.

### Analysis of cluster lifetimes

Cluster lifetimes were manually analyzed using FIJI. Deconvolved TIFF hyperstacks for individual nuclei were loaded on FIJI. Once a cluster was identified, it was tracked till it disappeared. The following criteria was used for tracking: a cluster was tracked as long as it remained within 600 nm (2 z-slices at 300 nm per slice) either above or below

the z position in the current frame. The start position, start time, end position, end time and total length of the movie were all recorded for each cluster. The normalized lifetime was calculated as the ratio of the total length of the cluster to the total length of the corresponding movie. A minimum of 5 nuclei were stochastically selected per embryo and a maximum of 8 clusters were sampled per nucleus. HLBs were excluded from this analysis except in nc11 where there are no clear, distinguishable HLBs.

### Simulations

Python simulations were designed to model the visibility of an RNAPII cluster at an active gene under a microscope. In the simulation, RNAPII molecules loaded onto a gene with a probability $k_{on}$, dissociated with a probability $k_{off}$, and elongated at a rate $r_{elongation}$ during periods when the promoter was in the ON state ($t_{on}$, set to 15 min to reflect bursts in nc14). The nucleus diameter was set to 5 μm (to reflect the nucleus diameter in nc14) and the RNAPII molecule diameter to 15 nm. The diameter around the active gene was set to 1 μm, consistent with the value used for our reporter gene analysis. Each molecule was convolved with a point spread function with a full width at half maximum set to 400 nm to mimic the optical resolution of the lattice light-sheet microscope. The total number of RNAPII molecules in a nucleus has been estimated at ~50,000 in nc14[62], and we scaled this down to reflect the number of molecules expected in a 3-slice max-projection in z at the center of the nucleus. The elongation rate was set to 2.5 kb/min[63] and the gene length was set to 2.8 kb (the length of or MS2 array and *yellow* reporter construct). The simulation generated one data point every 0.1 s, but to match our reporter imaging data, we subsampled the data every 9 s. Each output included both a "convolved" microscope image and an RNAPII intensity trace, representing the combined signal of transcribing molecules and background RNAPII in the active gene area. The resulting traces were then smoothed to match our experimental imaging analysis. The loading rates and burst durations were calculated using the same approach.

### Statistical analysis

All statistical analyses were performed using Python scripts. For each figure panel, the exact sample size (*n*-value) is reported in the figure legends. For the volumetric imaging data, the *n*-values refer to the number of independent embryos unless indicated otherwise. For single-molecule tracking, the number of trajectories obtained, independent fields of views, nuclei and embryos are all specified in the captions. Measurements from different embryos are considered as independent biological replicates. Normality of the data was assessed using the Shapiro-Wilk test where applicable. Statistical comparisons between conditions were performed using the non-parametric Mann-Whitney U-test (two-sided), unless otherwise specified. For compositional datasets (e.g., bound, intermediate, and fast RNAPII fractions), data were analyzed using a compositional data analysis (CoDA) decomposition framework as described in detail in the Methods section above. For all the plots, the central tendency is represented by the mean, and the variation is represented by the standard deviation unless otherwise noted. In cases where fractional values were derived, error bars were calculated using standard propagation of error as described earlier.

### Reporting summary

Further information on research design is available in the Nature Portfolio Reporting Summary linked to this article.

## Data availability

Source data are provided with this paper. Compiled data and code are available on Zeno https://doi.org/10.5281/zenodo.18233463.

## Code availability

Custom code used for analysis is available on Zenodo: https://doi.org/10.5281/zenodo.18233463.

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

## Acknowledgements

We thank Michael Stadler for initial discussions regarding this project and for providing critical fly lines. We thank Kaeli Mathias for help with Western blot analysis, Siewert Hugelier and Melike Lakadamyali for discussions on Voronoi clustering analysis, Priya Sivaramakrishnan for access to their lab confocal microscope (ZEISS LSM 900), and Nihanth Pinnaka for curating movies for the sister chromatid analysis. Additionally we thank all members of the Mir lab for insightful discussion and critical feedback and helpful discussions. We thank the CDB Microscopy Core (RRID SCR_022373) for the use of their confocal microscope (ZEISS LSM 880). The work was supported by National Institutes of Health grant DP2HD108775 (M.M.), Margaret Q Landenberger Foundation (M.M.), Howard Hughes Medical Institute, Freeman Hrabowski Scholars Program (M.M), and National Science Foundation Graduate Research Fellowship grant DGE-2236662 (to S.F. and E.H.).

## Author contributions

M.M. conceived and supervised the study. M.M. and A.M. constructed the light-sheet microscope and established single-molecule tracking. A.M. performed and analyzed single-molecule tracking and volumetric data with the assistance of K.S and M.K. K.S and R.D.C. optimized and performed embryo injections. S.F. developed software for cluster segmentation and assisted with analysis. M.K developed analysis software and simulations and analyzed live transcription imaging data. P.R. performed western blot analysis and generated key reagents. G.H generated fly lines. Y.I.H. performed cloning, EUTP microinjections, and generated fly lines. M.K and Y.I.H performed immunofluorescence and EUTP staining and confocal imaging. A.M., K.S., M.K., and M.M. interpreted the data, prepared figures, and wrote the initial manuscript. All authors contributed to review and editing. M.K. and K.S. are equally contributing co-second authors.

## Competing interests

The authors declare no competing interests.
