## [Transparent Peer Review file · Nature Communications]

A single cluster of RNA Polymerase II molecules is stably associated with active genes

Corresponding Author: Dr Mustafa Mir

Version 0:

Reviewer comments:

Reviewer #1

(Remarks to the Author)

The manuscript by Mukherjee et al. describes the use of lattice light-sheet microscopy to image RNA Polymerase 2 (RNAP2) clusters and transcription of the gene hunchback (hb) in *Drosophila* embryo during zygotic genome activation.

The role of condensates and/or hubs, RNAP2 at the promoter, or transcription factors at the enhancers, remains unclear a decade after transcriptional condensates came to the forefront. Live-cell imaging is uniquely poised to answer the minutia of this question, and the present study uses lattice light-sheet microscopy, an innovative method for imaging thick samples, which allows the authors to study living embryos of *Drosophila* at the period of zygotic genome activation (ZGA). ZGA is a great model to address these questions, as it marks the transition to a developmental stage with significantly higher transcription rates across the genome thus allowing to separate the chicken and the egg for many questions of transcription control. As such this study is of significant interest to the community. The authors provide a thorough examination of the relationship of how POL2 single molecule dynamics and clusters are affected during the ZGA transition, showing that transcriptional initiation is needed for the formation of RNAP2 clusters. Interestingly the authors' results help resolving some conflictual information in the literature, highlighting that clusters observed upon ZGA are mostly formed by elongating RNAP2, and yet elongation itself shortens the lifetime of these clusters. Finally, the simultaneous imaging of RNAP2 and nascent transcription at the hb gene, show that polymerase clustering and transcription are strongly correlated, suggesting that PolII clustering can last for the whole duration of time in which a gene is transcribed - in partial disagreement with previous findings suggesting that RNAP2 clusters mark the transcription initiation step.

The paper is well written, the methodology generally appears rigorous, and the findings are relevant and important. However, we think that some statements in the abstract and in the discussion should be revised, since the paper focuses on the relationship between RNAP2 clusters and the transcription of one single gene, and it is unclear whether the same results would hold if transcription at other genes would be tested. Further we have some specific comments on some of the analysis methods used.

1. The finding that RNAP2 clusters mark the whole duration of the hb transcriptional burst is in conflict with previous findings at other genes (Cho et al, 2022). The authors suggest that this difference might be due to different analysis procedures. While this is possible, since the analysis in the current manuscript is more precise and conceptually correct, it would be interesting to show what would happen if the authors would try to replicate Cho et al. analysis on their data on hb. Another possibility is that RNAP2 clustering could mark different phases of the transcription cycle for different genes, for example depending on chromatin topology, length of genes, transcription rates etc. For this reason, we would suggest the authors be more cautious in discussing their findings in general terms.

2. Similarly, the authors discuss the discrepancy between number of RNAP2 clusters observed and the number of activated genes upon ZGA: they provide two potential explanations (a) most RNAP2 clusters are composed by few polymerases and therefore they cannot be visualized (b) multiple genes are recruited to a single cluster and co-burst there together. The authors dismiss this second possibility as unlikely, given the large discrepancy between active genes and number of clusters. Yet, the most likely possibility is that both small RNAP2 clusters and co-recruitment of multiple genes at single clusters could coexist. According to this I believe that it cannot be concluded that the presented data could only be consistent with the "RNAP cluster transcribes one single gene" model.

3. In page 2 (Lines 89-91), the authors comment that the 1.4x increase in RNAP2 bound fraction observed at nc14, only partially reflects the increase in transcription (reportedly >2x) at this stage, and they start from this discrepancy to perform perturbation experiments to dissect between non-specifically engaged, initiating and elongating polymerases. This chain of thought seems a bit misleading: unless we missed it, the reported increase in transcription in the provided references relates to the number of genes that are transcribed rather than the amount of nascent transcripts (that should correlate with the amount of PolII actively elongating). A simpler (and in our opinion better) way to frame the perturbation experiments would be to just state that bound fractions measured by SMT report for a mixture of molecules that might be in different states, with confounding effect from non-specifically bound molecules.

4. page 6, line 218: “The data suggest that cluster lifetime may not be driven by cycle length but rather possibly the transcriptional cycle”.

We are unsure that this is the complete answer. Clearly for nc 11 and 12, the clusters’ lifetime is capped by mitosis. This could be understood as a manifestation of those aborted transcripts evidenced in Kwasnieski et al., 2019. Once the cell cycle is long enough to not be the limiting factor, other mechanisms can start becoming preponderant.

5. Statistical analysis of SMT fractions. The authors, like many others in the field, have used a Mann-Whitney U test to compare bound, intermediate and fast fractions across different conditions. However, this is formally inappropriate. As such fractions are compositional data, the assumption of independence is violated (bound + intermediate + fast =1). A helpful review on compositional data analysis can be found here: <https://doi.org/10.1146/annurev-statistics-042720-124436>. Maybe the authors could get inspiration from single-cell transcriptomics studies that frequently deal with compositional statistics. A possible method would be to first use centered log-ratio, then some multivariate hypothesis testing (MANOVA, regression, ...). While it is likely that the conclusion of the paper would not change, this would help the field in moving towards more correct statistical analysis.

6. Related to this. Figure 2e does not report errorbars for the fractions of elongating, initiating and non-specific binding quota of RNAP2, despite these could possibly be derived from the errors reported in fig 2d. Would the observed differences in elongating RNAP2 be significant between nc13 and nc14?

7. page 6, line 218: “The data suggest that cluster lifetime may not be driven by cycle length but rather possibly the transcriptional cycle”.

We are unsure that this is the complete answer. Clearly for nc 11 and 12, the clusters’ lifetime is capped by mitosis. This could be understood as a manifestation of those aborted transcripts evidenced in Kwasnieski et al., 2019. Once the cell cycle is long enough to not be the limiting factor, other mechanisms can start becoming preponderant.

8. Related to the previous point, it is interesting that the distribution of cluster dwell times shown in fig. S3.3 display peaks. While the peak at long times is probably due to the cut-off imposed by the cell cycle, is the early peak (observed for example at nc13) caused by some technicality? Or, maybe does it reflect the non-equilibrium nature of RNAP2 clusters?

9. Fig 4b. Why are the bound fractions reported in the control regions significantly lower than the ones outside the clusters. Shouldn't the first be just a subsampling of the second?

10. Figure 5d-e. We find it interesting that, at nc13 the cross-correlation plots appear asymmetric, with a second peak at ~+10min – also visible from the single cell cross correlation. Where does it come from? It might be useful to plot the autocorrelation of each of the signals, display whether it is the transcription or the clustering that reform 10 minutes after the cobursting. This would point towards some memory/refractory effect.

Minor Points:

- At page 3, line 118, the authors discuss using the MS2 reporter system for the first time, but they do not cite what gene it is marking until much later.
- Figure S.S1b. The label for eGFP-RBP1 is misspelled.

Reviewer #2

(Remarks to the Author)

Reviewer #3

(Remarks to the Author)

The manuscript of Mukherjee and colleagues examines behavior of Pol II molecules and clusters during *Drosophila* zygotic genome activation. They show that Pol II clusters form prior to transcription, and that these initial clusters depend on transcription initiation. After ZGA, the clusters shift towards elongating pol II. The findings are interesting to the gene

regulation and clustering communities, and highlight how cluster composition can shift over time. The experiments are performed thoroughly, but can be explained better at several places. I also have some alternative thoughts about the interpretation. Overall, these issues should be fixable with different representation and textual adjustments, which should make the manuscript suitable for publication.

Major comments:

Fig. S1.1. Based on these Western blots, it is not obvious that Pol II levels do not change. To me, the mEos-band appears less bright. Please quantify this (in multiple replicates)? Also, why is RPB1-Ser5P antibody used? To my understanding, this antibody only recognized initiating pol II.
Or alternatively, can the authors use quantitative Mass Spectrometry to verify this?

Fig 2E displays the fraction total bound Pol II that is either elongating, initiating or non-specifically bound, with the total scaled to total bound fraction. The current representation suggests that nr of initiating and non-specifically bound molecules decrease in nc14, but because the fraction bound is larger in nc14 than in nc13, these fraction are actually similar (if normalized to total Pol II). Only the elongating fraction actually increases. Please adjust and plot the fraction bound as in Fig 2d, and split that into the different fractions.

Fig 3 and S3 show deconvolved images, which enhance any spots/clusters. Can the authors also include raw images, so the reader can judge the clusters in unprocessed images?

"The data suggest that cluster lifetime may not be driven by cycle length but rather possibly the transcriptional cycle." This is mostly true after NC12. In those cases (NC13 and NC14), it may be better to represent the absolute lifetime, rather than the one normalized by interphase length. Also please adjust the concluding sentence: "but their normalized lifetimes shorten as transcriptional activity increases.", this is misleading because the cluster lifetime stays the same between nc13 and 14 (Fig S3.3a), it is the G1 phase that is prolonged.

Blocking elongation results in increase of cluster lifetime, from which the authors conclude that elongation dissolves cluster. Another explanation is that release of transcripts (after termination) dissolves clusters, but by blocking elongation, Pol II never reaches the end, resulting in Pol II staying bound to the locus and clusters getting a longer lifetime. Perhaps this is what is what the authors mean, but the way it is stated now suggests that they think the act of elongation (dNTP addition) dissolves Pol II clusters. The conclusion in the abstract should be adjusted accordingly: "cluster lifetimes are reduced upon transcription elongation".

Based on the discussion, I have the impression that the short lived fraction is thought to be elongation Pol II and the long-lived fraction are initiating clustered Pol II in the vicinity? Perhaps an earlier explanation of these fractions will help the reader with interpreting the data. I was very confused with this section (and how it relates to the sections after that) until I read the discussion.

Also, how do the timescales compare? The authors conclude that the clusters stay associated with the gene throughout the burst, but the crosscorrelation drops to zero correlation at 5 minutes. What is estimated burst duration/elongation time? And does this agree with the crosscorrelation timescales? For better interpretation, please include more time traces and also the autocorrelations, so the timescales can be compared. A video would also help.
Regarding these timescales. If the initiating clusters have longer lifetime, I would expect that clusters associate before transcription (elongation) starts. Why is this not visible at active genes?

Related, a large burst of Pol II can also result in multiple Pol II molecules in close proximity (i.e. a cluster). In that case, you also expect that the Pol II cluster intensity is correlated with the RNA signal Can the authors comment whether the Pol II they are detecting during the burst could simply be a burst, without invoking non-DNA bound or clustered molecules? If so, it is a bit confusing to call them clusters, since literature often uses this term in the context of "biomolecular condensates" where many molecules are non-DNA bound.

I am a bit surprised that the reporter gene associates with a cluster, given that there are so few clusters. Is this reporter gene particularly highly transcribed? A discussion on this would be useful.

In addition, given the density of clusters and the fact that the timewindow of transcriptional activity is so short (because of the G1 duration), where any active genes will be active at the same time, I wonder how specific this cluster association and correlation is. I am missing a negative control. Can they show that any other (negative control) gene or random position does not display similar cluster correlation by chance? Or can they show that these clusters only associate with (and correlate with) highly transcribed genes?

Section "RNAPII clusters are composed of elongating and kinetically trapped molecules after ZGA" All the fold changes are very confusing. Since the figures do not show the stated fold changes, it is sometimes unclear what is compared to what. For example: "In sharp contrast, in vehicle injected embryos, we found a 1.4-fold increase in the bound population", is this inside, outside/in control regions? Please be clearer what is compared and perhaps add the fold changes to the figures. Also, please include errors in these fold-changes.

Conceptually, I am also a bit confused why the outside and control are so different?

"A decomposition of this bound population as done above (Fig. 2e) leads to an estimate of a nearly 6-fold increase in the elongating population of RNAPII molecules within clusters from nc13 to nc14 (Fig. 4b)." Where does 6x come from? Again, please propagate the errors here. Does this calculation assume equal elongating/initiating/unspecific bound fraction in the

bound fraction of clusters versus outside clusters? I recommend rewriting this sentence and carefully stating the assumptions of this calculation (and explaining it)?

Minor comments:

"mean long-lived cluster lifetime is 6 ± 2 minutes in both nc13 and 7 ± 1 minutes in nc14." remove "both"

Legend Fig 3(a) and S3.1 should include more information what is displayed. From main text, I assume this is images of PolII-eGFP?

Title section "RNAPII clusters are composed of elongating and kinetically trapped molecules after ZGA" What does "kinetically trapped molecules" mean? Please rephrase

Fig 6 show elongating Pol II (producing transcripts, positioned after TSS) that are colored as initiating pol II. Please correct.

Reviewer #4

(Remarks to the Author)

In this study by Apratim et al., the authors use single-molecule tracking and lattice light sheet microscopy to characterize the kinetics of RNAPII before (nc13) and during ZGA (nc14) in *Drosophila*, showing an increase in the bound fraction of RNAPII. By using drugs to inhibit either initiation or elongation, they further dissect the bound fraction into initiating, elongating, and non-specific binding populations, and examine how this affects cluster density and lifetime of RNAPII clusters. They demonstrate an increase in the elongating RNAPII fraction within clusters from nc13 to nc14, and an increase in anisotropy of free RNAPII in clusters. Finally, they show the co-localization of an RNAPII cluster with a hunchback (hb) reporter gene, and that the cluster intensity positively correlates with the burst intensity of the hb gene.

Overall, this study provides an excellent demonstration of the RNAPII functional states in RNAPII clusters at different developmental stages and opens further questions in the field regarding the molecular control of these cluster dynamics and gene regulation. I believe this study has a solid foundation, and I have no doubt about the model the authors suggest, which is convincing. However, there are several technical areas that require revision:

1. Abstract: "We show that single clusters are stably associated with active gene loci during transcription." The authors only demonstrate this with one gene locus (hb).

2. The authors used saSPT for their trajectory analysis and attributed the intermediate state likely as confined molecules. This is a misconception: i) In the standard saSPT pipeline, all array fitting is based on Brownian motion. It is conceptually incorrect to classify molecules with lower fitted diffusion coefficients as confined, although they could be related. Confinement should be justified by anisotropy or other models such as radius of confinement. ii) In their methods, 100 diffusion coefficients were used for fitting. In the original saSPT paper, they emphasized selecting a parameter grid with spacing fine enough to avoid discretization artifacts. How did the authors determine the spacing? And how does changing that parameter affect the intermediate population? That should be further addressed and clarified. iii) The intermediate population could alternatively be explained by freely diffusing populations containing molecular species with different molecular weights. For example, the fast population could be RPB1 alone, while the intermediate could be the full RNAPII complex (or complex with TFIIIF), or the fast population could include truncated protein products. To show whether there are truncated products, Fig. S1.1 shows a Western blot, but it is not a full blot and only uses CTD Ser5P antibody, which is unusual. I think a pan-RPB1 antibody should be used. Fig. S1 also needs quantification to show similar expression levels. Overall, additional experiments and analysis are required to clarify the fast and intermediate populations. iv) For all saSPT plots, it is important to label the y-axis as "posterior probability" to inform readers unfamiliar with the technique that it is a fitted distribution.

3. Given the high concentration of RNAPII at the HLB, did the SMT analyses exclude trajectories from the HLB? Fig. 4 seems to exclude HLB data, but what about other SMT data? If not, how might the properties of RNAPII at HLB skew the SMT data at different conditions? This should be clarified.

4. When quantifying changes in bound fraction (e.g., Fig. 1d, 2d), it is important to consider changes in the number of RNAPII molecules per cell. In Fig. S2.1d, nc13 appears to have higher nuclear intensity of RNAPII than nc14. If the authors incorporate this into their calculations, how would the amount of bound RNAPII change, and how would that affect their biological interpretation?

5. I appreciate the approach of using drugs to dissect initiation, elongation, and non-specific binding. However, further experiments are needed to make the quantification convincing. The claim regarding non-specific binding does not seem robust. As shown in Fig. 3, the drug is not effective at HLB, indicating that the drugs could have gene-specific effects. To claim non-specific binding, the authors should at least conduct a dose-dependent assay to demonstrate they are using sufficient drug concentrations to inhibit almost all initiation or elongation events (shown by no further change in bound fraction) at non-HLB loci. Many claims about RNAPII states in clusters across developmental stages should also be validated using CTD Ser2P and Ser5P antibodies in fixed cells, as the drug assays do not provide direct evidence.

Additionally, how do triptolide and α -amanitin affect paused RNAPII? It should be addressed experimentally or at least discussed in the manuscript.

6. Regarding Fig. 3e, can the authors explain why α -amanitin causes a decrease in cluster density while the remaining clusters seem to be becoming larger? Do the clusters fuse? Can the authors examine the differences in the size of the clusters in different conditions for Fig. 3e?

7. The authors show that α -amanitin increases long-lived cluster lifetime in Fig. 3g,h. I would expect to see that reflected in Fig. 4a, but this does not appear to be the case. Presumably, if the "bound trajectories" were longer or appeared more frequently in the clusters, they should be prominent in the trajectory plot in Fig. 4a.

8. Regarding anisotropy: From Fig. S4.2, H2B trajectories in nc14 were used to determine the distance threshold. Can the authors also do the same analysis with H2B trajectories in nc13? If the two developmental stages exhibit distinct transcriptional activity, I would expect chromatin dynamics and H2B mobility to differ, potentially requiring different distance thresholds. Second, if I understand correctly, for Fig. 4, the "RNAPII cluster" is not a cluster at a given time but is defined by the average position of trajectories over an 80-second period. This means that if RNAPII clusters move during this period (which they certainly do, as shown in Fig. 3 and Fig. S3.1 on a 10-second scale), then the analysis area represents the space explored by RNAPII clusters over time, not at a specific moment. If cluster mobility differs between nc13, nc14, or after α -amanitin treatment, the anisotropy values would be confounded. I understand that the SMT was done at a very fast time scale (10ms/frame), but such movement of clusters should not be overlooked. This technical limitation could explain the modest 1.3-fold change, which might not be biologically meaningful.

9. Line 252 reports a "6-fold increase in the elongating population of RNAPII within clusters from nc13 to nc14" which is a key result to me. It would be helpful to show the underlying calculation, perhaps in a format similar to Fig. 2e, to clarify how this value was derived.

Version 1:

Reviewer comments:

Reviewer #1

(Remarks to the Author)

I would like to thank the authors for the thorough review of their manuscript. All my comments are now satisfactorily addressed.

(Remarks on code availability)

Reviewer #3

(Remarks to the Author)

The revision of Mukherjee et al. is much improved and I appreciate the rephrasing and additional analysis the authors have performed. I only have two remaining minor comments. If properly addressed, I fully support publication of this manuscript in Nature Communications.

-The authors addressed my comments of Fig 2e only partially. I appreciate the changes the authors made to this figure and it helps to have the nc13 and ncn14 next to each other for comparison. However, I still think it is confusing to normalize to the total bound Pol II, as this is changing between the conditions. This also does not match the description in the text: "This decomposition shows that the elongating fraction of RNAPII increases by $\sim 1.8 \pm 0.5$ -fold from nc13 to nc14", whereas figure 2e only shows a fold change of 1.2, from 0.45 to 0.55. Can the authors normalize this fraction to the total pol II, so the fraction represents the actual fraction of Pol II in the cells (rather than a fraction of a bound fraction)?

-In several instances, the authors mention "nascent transcription". Since transcription is always nascent, I suggest the authors use either "nascent transcripts" or "transcription".

(Remarks on code availability)

Reviewer #4

(Remarks to the Author)

The authors have done substantial work to address my concerns. The addition of analysis for three more genes significantly strengthens the study. Overall, the main conclusions are justified. However, I have three remaining points that require clarification. I do not request additional experiments, but the authors should address the following issues:

1. Regarding the statement "intermediate, which may reflect a heterogeneous population of molecules including those transiently confined, diffusing in complex with co-factors, truncated products with different molecular weights": In my original comment, I meant that the FAST population might include truncated products, not the intermediate population, as truncated products are expected to diffuse with a larger, not a smaller, diffusion coefficient.

2. Fig. R3 appears to contain errors. The authors state 200 diffusion coefficients, which I assume equals 200 grid spacings. However, Fig. R3 indicates 150 for grid spacing. The authors also mention this is from nc14, but by comparing with Fig. 1c, the data appears to be from nc13 instead. Please clarify.

3. The pan-RPB1 Western blot in Fig. S1.1 raises concerns about data interpretation. The error bars are quite large, with 0.4-fold differences observed for some replicates in panel e. The representative blot shown in panel d appears much stronger in eGFP-RPB1 than the quantified average suggests. The lack of statistical significance in panel e does not necessarily mean the samples are equivalent or similar to the untagged control, but rather reflects high variability in the data. To provide transparency, please include all four replicate Western blots in the supplement. Additionally, please clarify whether the quantification in panel e is based on the pan-RPB1 antibody or the Ser5 antibody.

(Remarks on code availability)

Revision document

We thank the reviewers for their thorough review of our manuscript, appreciation for the rigor of our data, and excitement regarding the broader impact of our findings. We have addressed all reviewer comments in our revised manuscript. Specifically, to add to our key finding of stable RNA Polymerase II (RNAPII) association with a single gene we have now quantified cluster interactions with three additional genes. We have also complemented these experiments with a simulation of cluster formation kinetics that elucidates which parameters of a transcription burst lead to an RNAPII cluster that is detectable over the background by lattice light-sheet microscopy. Finally, we have rearranged the data and figures as requested and clarified both figures and text as suggested. **Point-by-point responses and changes are detailed below in bold text.**

Reviewer #1 (Remarks to the Author):

The manuscript by Mukherjee et al. describes the use of lattice light-sheet microscopy to image RNA Polymerase 2 (RNAP2) clusters and transcription of the gene hunchback (hb) in *Drosophila* embryo during zygotic genome activation.

The role of condensates and/or hubs, RNAP2 at the promoter, or transcription factors at the enhancers, remains unclear a decade after transcriptional condensates came to the forefront. Live-cell imaging is uniquely poised to answer the minutia of this question, and the present study uses lattice light-sheet microscopy, an innovative method for imaging thick samples, which allows the authors to study living embryos of *Drosophila* at the period of zygotic genome activation (ZGA). ZGA is a great model to address these questions, as it marks the transition to a developmental stage with significantly higher transcription rates across the genome thus allowing to separate the chicken and the egg for many questions of transcription control. As such this study is of significant interest to the community. The authors provide a thorough examination of the relationship of how POL2 single molecule dynamics and clusters are affected during the ZGA transition, showing that transcriptional initiation is needed for the formation of RNAP2 clusters. Interestingly the authors' results help resolving some conflictual information in the literature, highlighting that clusters observed upon ZGA are mostly formed by elongating RNAP2, and yet elongation itself shortens the lifetime of these clusters. Finally, the simultaneous imaging of RNAP2 and nascent transcription at the hb gene, show that polymerase clustering and transcription are strongly correlated, suggesting that PolII clustering can last for the whole duration of time in which a gene is transcribed - in partial disagreement with previous findings suggesting that RNAP2 clusters mark the transcription initiation step.

The paper is well written, the methodology generally appears rigorous, and the findings are relevant and important. However, we think that some statements in the abstract and in the discussion should be revised, since the paper focuses on the relationship between RNAP2 clusters and the transcription of one single gene, and it is unclear whether the same results would hold if transcription at other genes would be tested. Further we have some specific comments on some of the analysis methods used.

1. The finding that RNAP2 clusters mark the whole duration of the hb transcriptional burst is in conflict with previous findings at other genes (Cho et al, 2022). The authors suggest that this difference might be due to different analysis procedures. While this is possible, since the analysis in the current manuscript is more precise and conceptually correct, it would be interesting to show what would happen if the authors would try to replicate Cho et al. analysis on their data on hb. Another possibility is that RNAP2 clustering could mark different phases of the transcription cycle for different genes, for example depending on chromatin topology, length of genes, transcription rates etc. For this reason, we would suggest the authors be more cautious in discussing their findings in general terms.

We agree that our data shows conflicting results compared to Cho et al., 2022 which shows the RNAPII signal peaking prior to the MS2 signal. While we also see clusters in the nucleus before the MS2 spot becomes visible, we cannot say whether a specific cluster is present at the MS2 locus before the spot becomes visible because accurately interpolating the position of the MS2 spot prior to it turning on is not possible. We attempted to replicate the Cho et al. analysis on our *hb* data and found a large degree of heterogeneity in when mean RNAPII intensity in all clusters peaks relative to the MS2 intensity - i.e., we did not see a consistent peak of the RNAPII signal before the MS2 signal. This could be due to segmentation differences (they segment on maximum-projected images while our cluster segmentation is done in three dimensions). Another possibility is nuclear cycle differences since Cho et al. focus on nc12 where transcription levels are low vs. nc13 and nc14 in our manuscript. We had addressed these differences in our Discussion:

Lines 445-452: “Our findings also contrast with previous observations in *Drosophila* embryos that reported RNAPII cluster intensities peak before transcription and decline well before burst termination, implying that only a minority of recruited polymerases enter into productive elongation¹⁷. This discrepancy may arise from methodological differences, as in this previous study¹⁷ the aggregate intensity of all RNAPII clusters in the nucleus was analyzed, whereas we directly measured the intensity of RNAPII clusters at an actively transcribing gene. Additionally, the study focused on nc12 where both transcription levels and the fraction of elongating polymerases is lower compared to nc13 and 14.”

We agree that different genes can show different RNAPII clustering based on properties such as gene length, enhancer environment, promoter on-times etc. To this end, we have carried out our analysis on three additional reporter lines (*eve*, *sna* and *sog*; Fig. 5a-d in the revised manuscript) and also developed a computational simulation where we can sweep these parameters and observe their effect on RNAPII cluster formation. We find that simply modulating the on-rate of RNAPII at the promoter is sufficient to mediate the appearance of a cluster over the background, and reflects parameters such as the loading rate of RNAPII and the burst duration (Fig. 5f-i in the revised manuscript).

2. Similarly, the authors discuss the discrepancy between number of RNAP2 clusters observed and the number of activated genes upon ZGA: they provide two potential explanations (a) most RNAP2 clusters are composed by few polymerases and therefore they cannot be visualized (b) multiple genes are recruited to a single cluster and co-burst there together. The authors dismiss this second possibility as unlikely, given the large discrepancy between active genes and number of clusters. Yet, the most likely possibility is that both small RNAP2 clusters and co-recruitment of multiple genes at single clusters could coexist. According to this I believe that it cannot be concluded that the presented data could only be consistent with the “RNAP cluster transcribes one single gene” model.

We agree with the reviewer that our data cannot definitively rule out that multiple genes are transcribed at a single RNAPII cluster. To further address this, we imaged a MS2 reporter construct post-replication, where we observed two separate MS2 spots on sister

chromatids (Fig. 5e). In these experiments, we consistently observed separate RNAPII clusters associated with each MS2 spot when the sister chromatids transiently separate to enough distances such that they can be resolved, rather than a single shared cluster bridging both loci.

In parallel, we performed simulations (Fig. 5f-i) demonstrating that, for a fixed gene length, the visibility of an RNAPII cluster is mediated simply by the RNAPII loading rate and transcription burst duration. Thus, for active genes with low loading rate or short bursts, RNAPII can still be locally enriched even though the enrichment may not surpass the background sufficiently to be detected as a discrete cluster. Taken together, our data favor a predominantly one-cluster-per-gene model, while recognizing that clusters below the detection threshold exist and that instances of multi-gene co-recruitment cannot be fully excluded. We have included this discussion in the revised manuscript (lines 422-438):

“The strong correlation between mRNA production at a single active gene and RNAPII cluster intensity suggests that each cluster could be associated with a single gene. Yet, despite the increase in transcriptional activity in nc14, the nuclear cluster density does not increase compared to nc13 (Fig. 3c), and the number of countable clusters per nucleus is far lower than the number of transcribed genes. This discrepancy could reflect two non-mutually exclusive possibilities: (1) most genes fail to recruit sufficient polymerases at a time to form visible clusters above the nuclear background in live imaging, or (2) clusters occasionally contain multiple co-bursting genes². Consistent with the first possibility, our simulations (Fig. 5f-i) show that the number of RNAPII molecules engaged with the active locus is enough to mediate the detectability of an RNAPII cluster over the background. For endogenous genes, this number can be influenced not only by a variable k_{on} , but also by the length and on-time of the gene (Movie 13). Furthermore, when we imaged sister chromatids post replication each containing an MS2 reporter allele, we observed distinct RNAPII clusters at each locus rather than a shared RNAPII cluster (Fig. 5e). Still, we cannot completely exclude multiple genes within an RNAPII cluster as also supported by Hi-C studies in *Drosophila* embryos showing that RNAPII is enriched at the boundaries of Topologically Associated Domains (TAD) and mediates inter-TAD compaction during development³.”

3. In page 2 (Lines 89-91), the authors comment that the 1.4x increase in RNAP2 bound fraction observed at nc14, only partially reflects the increase in transcription (reportedly >2x) at this stage, and they start from this discrepancy to perform perturbation experiments to dissect between non-specifically engaged, initiating and elongating polymerases. This chain of thought seems a bit misleading: unless we missed it, the reported increase in transcription in the provided references relates to the number of genes that are transcribed rather than the amount of nascent transcripts (that should correlate with the amount of PolIII actively elongating). A simpler (and in our opinion better) way to frame the perturbation experiments would be to just state that bound fractions measured by SMT report for a mixture of molecules that might be in different states, with confounding effect from non-specifically bound molecules.

We agree that a simple way to motivate the perturbation experiments is to decompose the bound fraction further into the different components rather than discuss the >2x increase in the number of genes being activated during ZGA. Thus, we have removed the reference

to that phrase and have modified the text in the manuscript (lines 100-107) to reflect this as also shown below:

“Overall, RNAPII molecules transition, from fast and intermediate kinetic states, to bound, during ZGA, consistent with increasing transcriptional activity. However, our single-molecule localization error of ~30 nm is insufficient to distinguish between RNAPII molecules engaged in different steps of the transcription cycle, namely engagement, initiation, and elongation, all of which would appear as chromatin-bound in our data. To deconvolve this chromatin-bound population, we next turned to small-molecule inhibitors that selectively block specific stages of the transcription cycle⁴.”

Furthermore, we realize that beginning the results section by mentioning purely the increase in the number of genes being transcribed can be confusing. We have now added language in the manuscript emphasizing the overall increase in the number of transcripts during ZGA, which should be reflected in the single-molecule kinetics data (lines 80-85). These changes are also shown below:

“The major wave of ZGA in *Drosophila* embryos occurs in the 14th nuclear cleavage cycle (nc14) during which the amount of nascent transcription increases sharply along with the number of active genes. There are ~1,000 active genes before nc14, increasing to upwards of 3,500 genes in nc14, with estimates varying by the method of quantification⁵⁻⁹ (Fig. 1a). We reasoned that this increase in transcriptional activity should correspond to an increase in the chromatin bound fraction of RNAPII molecules when measured using single-molecule tracking.”

4. page 6, line 218: “The data suggest that cluster lifetime may not be driven by cycle length but rather possibly the transcriptional cycle”.

We are unsure that this is the complete answer. Clearly for nc 11 and 12, the clusters' lifetime is capped by mitosis. This could be understood as a manifestation of those aborted transcripts evidenced in Kwasnieski et al., 2019. Once the cell cycle is long enough to not be the limiting factor, other mechanisms can start becoming preponderant.

We agree with the reviewer that in the earlier cycles (nc11 and nc12), cluster lifetimes are likely capped by mitosis, consistent with the aborted transcripts described by Kwasnieski et al., 2019⁵. Once the cell cycle lengthens, however, other mechanisms can become predominant regulators of clustering kinetics. To highlight this distinction, we have revised Figure 3 to now present both lifetimes normalized to interphase length (Fig. 3d) and absolute lifetimes (Fig. 3e). The absolute lifetimes make clear that mitosis likely constrains cluster lifetimes in nc11–12, while in nc13–14, the duration of transcription bursts are likely setting lifetimes. We have also edited lines 230-245 in the Results section to (i) include a reference to the aborted transcripts reported by Kwasnieski, et al., 2019, and (ii) clarify that in later cycles, such as nc13 and nc14, transcription kinetics likely modulate clustering lifetimes:

“While the majority of clusters in the early cycles (nc11-12) persist through interphase, we found that cluster lifetimes, when normalized to interphase duration, shorten significantly as ZGA progresses (Fig. 3d). This reduction in lifetimes suggests that in nc11 and nc12, absolute cluster lifetimes (Fig. 3e) are constrained by the short interphase durations consistent with previous reports of frequently aborted transcription during these stages⁵. In contrast, during later cycles (nc13-14), absolute cluster lifetimes remain similar but are no longer capped by interphase duration, leading to shorter normalized values (Fig 3d,e). The distribution of cluster lifetimes are well described by a short and long-lived population (Fig. S3.2d, Movie 8). Strikingly, the fraction of long-lived clusters decreases from $64\pm 2\%$ in nc11 down to only $33\pm 3\%$ in nc14. The data suggest that cluster lifetime in nc13 and nc14, may not be limited by cycle length but rather depends on the rate of transcriptional activity, as supported by the concurrent rise in the elongating fraction of RNAPII (Fig. 2e). Consistent with this shift in lifetimes, immunofluorescence analysis shows that RNAPII molecules in clusters shift from being in mostly initiating states (Ser5P staining) during the early cycles (nc11-12), to largely elongating states (Ser2P staining) in nc14 (Fig. S3.3).”

5. Statistical analysis of SMT fractions. The authors, like many others in the field, have used a Mann-Whitney U test to compare bound, intermediate and fast fractions across different conditions. However, this is formally inappropriate. As such fractions are compositional data, the assumption of independence is violated ($\text{bound} + \text{intermediate} + \text{fast} = 1$). A helpful review on compositional data analysis can be found here: <https://doi.org/10.1146/annurev-statistics-042720-124436>. Maybe the authors could get inspiration from single-cell transcriptomics studies that frequently deal with compositional statistics. A possible method would be to first use centered log-ratio, then some multivariate hypothesis testing (MANOVA, regression, ...). While it is likely that the conclusion of the paper would not change, this would help the field in moving towards more correct statistical analysis.

We agree that compositional data analysis (CoDA) is an important and underutilized framework in the field for performing statistical analysis on SMT datasets. As the reviewer noted, bound, intermediate, and fast fractions derived from single-molecule tracking are compositional and constrained to sum to one, violating the independence assumptions required for standard statistical tests. To address this issue, in the revised manuscript, we performed a CoDA by first computing the geometric mean of the three fractions for each movie and then applying a centered log-ratio (CLR) transformation, as is common practice for such analysis. We subsequently used the Mann-Whitney U test to compare CLR-transformed values between conditions. This updated analysis is now reflected in Fig. 1 and Fig. 2, where we compare SMT fractions between nc13 and nc14 in the vehicle, α -amanitin, and Triptolide conditions. Importantly, the statistical significance of our findings remains unchanged after transformation. For data presented in Fig. 4, only the individual bound fraction is compared across conditions, and thus it is not subject to the compositional constraint.

These details have been added to the Methods in a new section entitled “Compositional data analysis (CoDA) of SMT fractions”.

6. Related to this. Figure 2e does not report error bars for the fractions of elongating, initiating and non-specific binding quota of RNAP2, despite these could possibly be derived from the errors reported in fig 2d. Would the observed differences in elongating RNAP2 be significant between nc13 and nc14?

We have now added error bars to the elongating, initiating, and non-specific RNAPII fractions in Fig. 2e by propagating the standard errors that were reported in Fig. 2d.

Our estimates of the increase in the elongating fraction are based on subtracting the mean bound fraction in α -amanitin-injected embryos (which reflects the non-specific and initiating populations) from the total bound fraction in vehicle-injected embryos. We have now added details about how this calculation is made in the Methods section under the section “Analysis of single-molecule tracking data.” In nc13, the elongating fraction is 0.46 ± 0.06 (95% CI: 0.34–0.58), while in nc14 it is 0.60 ± 0.07 (95% CI: 0.46–0.74), corresponding to a ~1.8-fold increase. Since these measurements come from different sets of embryos, it is not possible to calculate a “per-embryo” elongating fraction or its change across nuclear cycles. This precludes any formal statistical testing of the increase in the elongating fraction during ZGA, which is why we provide the confidence interval values for reference.

7. page 6, line 218: “The data suggest that cluster lifetime may not be driven by cycle length but rather possibly the transcriptional cycle”.

We are unsure that this is the complete answer. Clearly for nc 11 and 12, the clusters’ lifetime is capped by mitosis. This could be understood as a manifestation of those aborted transcripts evidenced in Kwasnieski et al., 2019. Once the cell cycle is long enough to not be the limiting factor, other mechanisms can start becoming preponderant.

We agree with the reviewer that in the earlier cycles (nc11 and nc12), cluster lifetimes are likely capped by mitosis, consistent with the aborted transcripts described by Kwasnieski et al., 2019⁵. Once the cell cycle lengthens, however, other mechanisms can become predominant regulators of clustering kinetics. To highlight this distinction, we have revised Figure 3 to now present both lifetimes normalized to interphase length (Fig. 3d) and absolute lifetimes (Fig. 3e). The absolute lifetimes make clear that mitosis likely constrains cluster lifetimes in nc11–12, while in nc13–14, the duration of transcription bursts are likely setting lifetimes. We have also edited lines 230-245 in the Results section to (i) include a reference to the aborted transcripts reported by Kwasnieski, et al., 2019, and (ii) clarify that in later cycles, such as nc13 and nc14, transcription kinetics likely modulate clustering lifetimes:

“While the majority of clusters in the early cycles (nc11-12) persist through interphase, we found that cluster lifetimes, when normalized to interphase duration, shorten significantly as ZGA progresses (Fig. 3d). This reduction in lifetimes suggests that in nc11 and nc12, absolute cluster lifetimes (Fig. 3e) are constrained by the short interphase durations consistent with previous reports of frequently aborted transcription during these stages⁵. In contrast, during later cycles (nc13-14), absolute cluster lifetimes remain similar but are no longer capped by interphase duration, leading to shorter normalized values (Fig 3d,e). The distribution of cluster lifetimes are well described by a short and long-lived population (Fig. S3.2d, Movie 8). Strikingly, the fraction of long-lived clusters decreases from $64 \pm 2\%$ in nc11 down to only $33 \pm 3\%$ in nc14. The data suggest that cluster lifetime in nc13 and nc14, may not be limited by cycle length but rather depends on the rate of transcriptional activity, as supported by the concurrent rise in the elongating fraction of RNAPII (Fig. 2e). Consistent with this shift in lifetimes, immunofluorescence analysis shows that RNAPII molecules in clusters shift from being in mostly initiating states (Ser5P staining) during the early cycles (nc11-12), to largely elongating states (Ser2P staining) in nc14 (Fig. S3.3).”

8. Related to the previous point, it is interesting that the distribution of cluster dwell times shown in fig. S3.3 display peaks. While the peak at long times is probably due to the cut-off imposed by the cell cycle, is the early peak (observed for example at nc13) caused by some technicality? Or, maybe does it reflect the non-equilibrium nature of RNAP2 clusters?

As shown in Figure S3.2d,e, the distribution of cluster dwell times in each nuclear cycle is well described by a two-component Gaussian mixture model, corresponding to a short- and long-lived population, as currently detailed in the Methods. The “early peak” noted by the reviewer in nc13 and nc14 is not a technical artifact but rather reflects the emergence of a distinct, increasingly dominant short-lived population in the latter nuclear cycles. This point is further underscored in Figure 3i, where the gray bars represent the fraction of short-lived clusters, which increases from $36 \pm 2\%$ in nc11 up to $67 \pm 3\%$ in nc14.

9. Fig 4b. Why are the bound fractions reported in the control regions significantly lower than the ones outside the clusters. Shouldn't the first be just a subsampling of the second?

We were initially defining 10 control spots for each cluster identified. With each spot having a radius of 200 nanometers, we were only sampling approximately 6% of the nuclear area being imaged. We now increased the number of control spots to 30, and obtained a bound fraction corresponding to the “outside the cluster” values. We have accordingly updated Fig. 4 in the main text. Additionally, we find that increasing the number of control spots any further does not change the bound fraction, suggesting that ~30 control spots is an appropriate number for this analysis (Fig. R1).

Fig. R1: Bound fraction of RNAPII trajectories inside control spots as a function of the number of control spots in vehicle injected embryos. Data points represent 23 individual fields of view with ~8-15 nuclei per field of view. Error bars show standard deviation.

10. Figure 5d-e. We find it interesting that, at nc13 the cross-correlation plots appear asymmetric, with a second peak at ~+10min – also visible from the single cell cross correlation. Where does it come from? It might be useful to plot the autocorrelation of each of the signals, display whether it is the transcription or the clustering that reform 10 minutes after the cobursting. This would point towards some memory/refractory effect.

We plotted autocorrelation functions for each of the signals as per the reviewers suggestion. We found no consistent, significant peak at ~+10 minutes and believe that the increase in cross-correlation near the edge of each trace was due to edge effects arising from a shorter signal being overlapped or due to RNAPII reloading for a second transcriptional burst. We use the correlation method “full” for `scipy.signal.correlate()`, which pads the traces to enable complete sliding of one trace along the other in time, and this padding can lead to increases in cross-correlation at extreme lags.

To improve the accuracy and functional relevance of our cross-correlations we have updated our analyses to look at individual bursts instead of the complete length of the MS2 spot tracks during a nuclear cycle. Again, we do not see a consistent spike in cross-correlation, but we do see the same edge effects in some cases due to small overlaps and padding. The autocorrelations for these individual bursts are shown in Fig. R2.

Fig. R2: Average (line plot) and individual (heatmap) autocorrelations for the eGFP-RPB1 (top) or MCP-mCh (bottom) intensities around the MS2 spot at each of the four genes in nc14, from three biological replicates.

Minor Points:

- At page 3, line 118, the authors discuss using the MS2 reporter system for the first time, but they do not cite what gene it is marking until much later.

We have now updated this to indicate that we looked at transcription of hunchback (*hb*) for the injection validation experiments.

- Figure S.S1b. The label for eGFP-RBP1 is misspelled.

We have corrected this typo.

Reviewer #2 (Remarks to the Author):

We thank the reviewer for their contributions to this review.

Reviewer #3 (Remarks to the Author):

The manuscript of Mukherjee and colleagues examines the behavior of Pol II molecules and clusters during *Drosophila* zygotic genome activation. They show that Pol II clusters form prior to transcription, and that these initial clusters depend on transcription initiation. After ZGA, the clusters shift towards elongating pol II. The findings are interesting to the gene regulation and clustering communities, and highlight how cluster composition can shift over time. The experiments are performed thoroughly, but can be explained better at several places. I also have some alternative thoughts about the interpretation. Overall, these issues should be fixable with different representation and textual adjustments, which should make the manuscript suitable for publication.

Major comments:

1. Fig. S1.1. Based on these Western blots, it is not obvious that Pol II levels do not change. To me, the mEos-band appears less bright. Please quantify this (in multiple replicates)? Also, why is RPB1-Ser5P antibody used? To my understanding, this antibody only recognized initiating pol II. Or alternatively, can the authors use quantitative Mass Spectrometry to verify this?

We initially performed the Western blot using an RPB1-Ser5P antibody because this was the only available antibody validated in *Drosophila* and we reasoned that it reported on the relevant chromatin bound fraction of RPB1. We have now repeated the Western blot analysis using a pan-RPB1 antibody across multiple replicates. Quantifying small changes in protein abundance is not ideal using western blots but we do agree that the mEos expression appears to be lower in some replicates, but this difference is not statistically significant. Both eGFP and mEos3.2 tagged fly lines and embryos are homozygous viable and have been propagated as such for more than a year with no loss of fecundity. We have added the following lines to the text:

Lines 88-89: This insertion is homozygous-viable over hundreds of generations of propagation and only homozygous embryos were imaged.

Lines 191-193: To examine RNAPII clustering during ZGA, we performed volumetric lattice light-sheet imaging in *Drosophila* embryos with endogenous RPB1 tagged with eGFP at its N-terminus at both alleles. (Fig. S1.1).

We have updated Fig. S1 to include quantification across multiple replicates of western blots and a representative full blot using a pan-RPB1 antibody is shown.

2. Fig 2E displays the fraction total bound Pol II that is either elongating, initiating or non-specifically bound, with the total scaled to total bound fraction. The current representation suggests that nr of initiating and non-specifically bound molecules decrease in nc14, but because the fraction bound is larger in nc14 than in nc13, these fraction are actually similar (if normalized to total Pol II). Only the elongating fraction actually increases. Please adjust and plot the fraction bound as in Fig 2d, and split that into the different fractions.

We agree that the original visualization in Fig. 2e could be misleading and have revised the figure accordingly. As suggested, we now present adjacent bars for nc13 and nc14, depicting the contributions of non-specific, initiating, and elongating RNAPII subpopulations to the total bound fraction (as in Fig. 2d). We have also added error bars using standard error propagation, addressing a similar comment from Reviewer 1 This updated representation clarifies that neither the non-specific nor initiating fractions (as a proportion of the total bound RNAPII) change significantly between nc13 and nc14, while the elongating fraction shows a marked increase. This supports the original interpretation (lines 181-186), which stated:

“Thus the bound fractions from triptolide and α -amanitin injected embryos can be used to estimate the fractions of initiating and non-specifically interacting molecules within the bound fraction in vehicle injected embryos. Using this fact we decomposed the bound fraction in vehicle injected embryos into initiating, elongating, and non-specifically interacting molecule components (Fig. 2e). This decomposition shows that the elongating fraction of RNAPII increases by $\sim 1.8 \pm 0.5$ -fold from nc13 to nc14.”

Furthermore, to make it clear to the reader how these individual fractions were estimated from the bound fraction values, we have now included a detailed description of this in the Methods section (lines 844-857). This section is also presented below:

“To estimate the relative contributions of non-specifically bound, initiating, and elongating RNAPII molecules within the total bound fraction, we used a subtraction-based decomposition approach. For each nuclear cycle, the total bound fraction in vehicle-injected embryos was normalized to 1, representing the sum of the three subpopulations. The bound fraction measured in triptolide-injected embryos was assigned to the non-specific component. The difference between the bound fractions in α -amanitin and triptolide-injected embryos was attributed to the initiating component. The elongating fraction was thus calculated as the remainder, by subtracting the α -amanitin bound fraction from the total bound fraction in the vehicle-injected embryos. This analysis was performed separately for nc13 and nc14 to obtain the fraction change in the elongating population. Standard error propagation was used to estimate the error bars for each component. For the kinetics inside the clusters, we did not differentiate between the non-specific binding fraction and the initiating fraction for the fold-change estimation because there were minimal observable clusters left after triptolide injection.”

3. Fig 3 and S3 show deconvolved images, which enhance any spots/clusters. Can the authors also include raw images, so the reader can judge the clusters in unprocessed images?

We agree with the reviewer and have now updated Fig. S3.1 to include both raw (unprocessed) and deconvolved images for every snapshot in each nuclear cycle and each time point shown. The first time points for each nuclear cycle in this figure are used in Fig. 3.

4. "The data suggest that cluster lifetime may not be driven by cycle length but rather possibly the transcriptional cycle. " This is mostly true after NC12. In those cases (NC13 and NC14), it may be better to represent the absolute lifetime, rather than the one normalized by interphase length. Also please adjust the concluding sentence: "but their normalized lifetimes shorten as transcriptional activity increases.", this is misleading because the cluster lifetime stays the same between nc13 and 14 (Fig S3.3a), it is the G1 phase that is prolonged.

We agree that while the normalized cluster lifetimes showed a decreasing trend from nc11 to nc14, this pattern also reflects the increasing interphase duration. As suggested, we now present both the normalized lifetimes (Fig. 3d) and the absolute lifetimes (Fig. 3e). This makes it clear that the long-lived cluster population maintains relatively stable lifetimes between nc13 and nc14 (~7 minutes), while the more meaningful shift is the reduction in the fraction of long-lived clusters.

We have also revised the text in several places to reflect this distinction including the Abstract and the Introduction:

-lines 26-29 in the Abstract: "Using single-molecule tracking and lattice light-sheet microscopy, we find that RNAPII cluster formation depends on transcription initiation, and that cluster lifetimes depend on transcriptional activity when not constrained by interphase duration."

-lines 73-75 in the Introduction: "We find that RNAPII cluster formation depends on transcription initiation and that their lifetimes depend on transcriptional activity."

-lines 262-264 which now state: "Furthermore, clusters emerge well before the major wave of ZGA, and their properties change with an increase in transcriptional activity. While the absolute lifetimes of long-lived clusters stay relatively stable, their fraction decreases markedly as transcriptional activity increases."

-lines 397-401 which now state: "The increase in elongation within clusters is supported by an increase in Ser2 phosphorylation signal in RNAPII clusters (marking elongating RNAPII molecules; Fig. S3.2) in nc14 compared to early cycles and by the reversion of lifetimes and densities upon elongation inhibition (Fig. 3f-h)."

-lines 410-414 which now state: "The high correlation of local RNAPII intensity and transcriptional output we measure (Fig. 5) suggests that the shift toward shorter-lived clusters may simply reflect an increase in the number of RNAPII molecules being released into the gene body and then completing transcription, rather than higher-order regulatory mechanisms."

5. Blocking elongation results in increase of cluster lifetime, from which the authors conclude that elongation dissolves cluster. Another explanation is that release of transcripts (after termination) dissolves clusters, but by blocking elongation, Pol II never reaches the end, resulting in Pol II staying bound to the locus and clusters getting a longer lifetime. Perhaps this is what is what the authors mean, but the way it is stated now suggests that they think the act of elongation (dNTP addition) dissolves Pol II clusters. The conclusion in the abstract should be adjusted accordingly: "cluster lifetimes are reduced upon transcription elongation".

We agree that RNA-mediated effects on cluster stability have been proposed earlier which we have already cited in our Discussion section as shown below:

“Additionally, the dissolution of transcriptional condensates in response to increased transcriptional activity has been proposed to arise from changes in electrostatic forces from local RNA concentrations¹⁰. The high correlation of local RNAPII intensity and transcriptional output we measure (Fig. 5) suggests that the shift toward shorter-lived clusters may simply reflect an increase in the number of RNAPII molecules being released into the gene body and then completing transcription, rather than higher-order regulatory mechanisms. In this context, our use of the term “cluster” does not imply a specific assembly mechanism such as phase separation.”

As we highlight in this section, we interpret cluster visibility and lifetime primarily as a function of the number of RNAPII molecules at the active locus. When elongation is slowed by α -amanitin, polymerase molecules accumulate near the promoter and dwell longer, extending cluster lifetimes. Conversely, we interpret the loss of a visible cluster as occurring when engaged polymerases complete transcription and no new molecules are loaded, leading to the signal falling below the detection threshold. Our simulations (Fig. 5f-i) are able to recapitulate this detectability threshold by varying the RNAPII loading probability at the promoter (k_{on}), which in turn affects the RNAPII loading rate and burst duration. Accordingly, while termination or RNA-mediated effects may contribute, they are not required to explain our experimental data or simulations and we thus avoid attributing dissolution directly to termination.

6. Based on the discussion, I have the impression that the short lived fraction is thought to be elongation Pol II and the long-lived fraction are initiating clustered Pol II in the vicinity? Perhaps an earlier explanation of these fractions will help the reader with interpreting the data. I was very confused with this section (and how it relates to the sections after that) until I read the discussion.

We agree that clarification would be helpful regarding the interpretation of the short- and long-lived cluster populations. In the revised manuscript, we now provide an explanation earlier in the Results section (lines 256-259):

“While these results show that transcriptional inhibition shifts the cluster lifetime distribution, we do not interpret short- and long-lived clusters as strictly corresponding to elongating or initiating RNAPII respectively. Rather, cluster lifetimes likely reflect a mixture of transcriptional states”

We also note that our imaging approach does not offer the spatial resolution required to distinguish between initiating and elongating RNAPII within individual clusters. As such, we do not assign specific transcriptional states to clusters based solely on their lifetimes.

Our data show that earlier nuclear cycles (nc11-12) contain a higher proportion of long-lived clusters, whereas later cycles (nc13-14) exhibit a pronounced shift toward short-lived clusters. We interpret this shift as increased transcriptional activity in later cycles, where RNAPII molecules are more rapidly released into elongation. This interpretation is consistent with our α -amanitin experiments, in which inhibition of elongation produced a dramatic increase in both the fraction and the lifetime of long-lived clusters. To further contextualize this finding, our new simulations (Fig. 5f-i). suggest that simply the number of engaged RNAPII molecules at an active locus determines whether RNAPII enrichment appears as a visible cluster, underscoring that cluster lifetime alone cannot be used to assign a specific transcriptional state.

7. Also, how do the timescales compare? The authors conclude that the clusters stay associated with the gene throughout the burst, but the crosscorrelation drops to zero correlation at 5 minutes. What is estimated burst duration/elongation time? And does this agree with the crosscorrelation timescales? For better interpretation, please include more time traces and also the autocorrelations, so the timescales can be compared. A video would also help.

Regarding these timescales. If the initiating clusters have longer lifetime, I would expect that clusters associate before transcription (elongation) starts. Why is this not visible at active genes?

In order to better represent the timescales, we have updated our analyses to show individual bursts instead of the full length of time the MS2 spot is visible. We see that bursts, as called by a local minima detection, can range from under a minute up to ~18 minutes and hence are highly heterogeneous even within the same embryo for the same gene. The cross-correlation measures how correlated signals are in time and the lag times do not reflect the burst duration. For instance, if the RNAPII signal is shifted to the right by 5 minutes, it is possible that MCP has started to plateau, which would give a cross-correlation of 0. We cannot determine if an RNAPII cluster forms prior to transcription as the MS2 spot moves significantly between frames, making it difficult to accurately extrapolate its position.

We have added more time traces for all 4 of our reporter genes in Fig. S5.2. We have also added videos for all 4 reporter genes (Movie 11). Autocorrelations are plotted in Fig. R2.

8. Related, a large burst of Pol II can also result in multiple Pol II molecules in close proximity (i.e. a cluster). In that case, you also expect that the Pol II cluster intensity is correlated with the RNA signal. Can the authors comment whether the Pol II they are detecting during the burst could simply be a burst, without invoking non-DNA bound or clustered molecules? If so, it is a bit confusing to call them clusters, since literature often uses this term in the context of “biomolecular condensates” where many molecules are non-DNA bound.

I am a bit surprised that the reporter gene associates with a cluster, given that there are so few clusters. Is this reporter gene particularly highly transcribed? A discussion on this would be useful.

We agree that a transcriptional burst involving multiple elongating RNAPII molecules can result in a local enrichment at a gene locus, which may appear as a “cluster” in our

imaging. These clusters could comprise a combination of chromatin-bound RNAPII molecules as well as a non-bound or a free component, as highlighted by the analysis of trajectories within the clusters in Fig. 4 in the manuscript.

Importantly, our use of the term “cluster” is not meant to imply a specific physical mechanism such as phase separation. Rather, it reflects the local accumulation of RNAPII molecules at a gene locus. This is consistent with recent findings from Gaskill, et al., 2023, who investigated the transcription factor GAF in *Drosophila* embryos and showed that its nuclear foci, which visually resembled condensates, could be fully explained by high-affinity binding to clustered genomic loci alone, without invoking phase separation or non-specific self-interactions¹¹. Similarly, our simulations (Fig. 5f-i) demonstrate that the presence of a detectable RNAPII cluster at an active gene is determined simply by the number of engaged RNAPII molecules, even in the absence of mechanisms that would drive condensate formation. Together, these results suggest that visible clusters can emerge from a local accumulation of DNA-bound molecules alone. While we cannot exclude that non-DNA-bound interactions may contribute to clustering, our functional data (enrichment of elongating RNAPII, correlation with nascent transcription, and modulation by transcriptional inhibitors) strongly support the interpretation that these foci represent functional sites of transcription. We now clarify this in the Discussion (lines 410 - 419) as shown below:

“The high correlation of local RNAPII intensity and transcriptional output we measure (Fig. 5) suggests that the shift toward shorter-lived clusters may simply reflect an increase in the number of RNAPII molecules being released into the gene body and then completing transcription, rather than higher-order regulatory mechanisms. In this context, our use of the term “cluster” does not imply a specific assembly mechanism such as phase separation. Indeed, our simulations (Fig. 5) show that bursts of RNAPII loading and elongation alone can generate RNAPII enrichment at an active gene without invoking a biomolecular condensate model. This is consistent with recent findings on the transcription factor GAF, where foci that visually resemble condensates were shown to arise purely from high chromatin occupancy at satellite repeat clusters¹¹. ”

Regarding the question of the stable association of an RNAPII cluster with an active gene given the limited number of clusters observed in nc14, we refer back to our simulations that show that the presence of an observable cluster can be modulated simply by the polymerase loading rate and burst duration. Thus, there could be many genes that are transcribing in nc14 without a detectable RNAPII cluster as we note in our Discussion (lines 423-429):

“Yet, despite the increase in transcriptional activity in nc14, the nuclear cluster density does not increase compared to nc13 (Fig. 3c), and the number of countable clusters per nucleus is far lower than the number of transcribed genes. This discrepancy could reflect two non-mutually exclusive possibilities: (1) most genes fail to recruit sufficient

polymerases at a time to form visible clusters above the nuclear background in live imaging, or (2) clusters occasionally contain multiple co-bursting genes²."

9. In addition, given the density of clusters and the fact that the time window of transcriptional activity is so short (because of the G1 duration), where any active genes will be active at the same time, I wonder how specific this cluster association and correlation is. I am missing a negative control. Can they show that any other (negative control) gene or random position does not display similar cluster correlation by chance? Or can they show that these clusters only associate with (and correlate with) highly transcribed genes?

We have updated Fig. 5 to now show correlations with four reporter genes - *eve*, *hb*, *sna* and *sog*.

We agree about the importance of a negative control and to that effect we have done the following:

(1) normalized our cluster intensity to the nuclear background

(2) calculated cross-correlations between the MS2 spot and a random control spot in nuclei that exhibited high cross-correlations for all four genes. We observed that the correlations decreased significantly (Fig. S5.3) but did not drop to 0, which we believe is due to changes in the nuclear concentration of RNAPII at the beginning and end of a nuclear cycle.

(3) imaged a non-transcribing, inverted *Ubx* locus, marked using the ParB/ParS system and observed no long-term stable association of eGFP-RPB1 clusters with the locus (Fig. S5.1d).

10. Section "RNAPII clusters are composed of elongating and kinetically trapped molecules after ZGA" (i) All the fold changes are very confusing. Since the figures do not show the stated fold changes, it is sometimes unclear what is compared to what. For example: "In sharp contrast, in vehicle injected embryos, we found a 1.4-fold increase in the bound population", is this inside, outside/in control regions? Please be clearer what is compared and perhaps add the fold changes to the figures. (ii) Also, please include errors in these fold-changes.

(iii) Conceptually, I am also a bit confused why the outside and control are so different?

"A decomposition of this bound population as done above (Fig. 2e) leads to an estimate of a nearly 6-fold increase in the elongating population of RNAPII molecules within clusters from nc13 to nc14 (Fig. 4b)." (iv) Where does 6x come from? Again, please propagate the errors here. Does this calculation assume equal elongating/initiating/unspecific bound fraction in the bound fraction of clusters versus outside clusters? I recommend rewriting this sentence and carefully stating the assumptions of this calculation (and explaining it)?

The reviewer makes several different points here which we address in order:

(i) We agree with the reviewer that parsing apart all the fold changes might be difficult in this section. We have accordingly updated the language in our main text to make it explicitly clear what is being compared. Below is the updated language:

“By comparing RNAPII kinetics inside clusters versus the rest of the nucleoplasm (outside clusters), we found a 1.4 ± 0.4 -fold enrichment of bound RNAPII trajectories inside clusters relative to outside clusters in both nc13 and nc14 in vehicle-injected embryos and a 2.5 ± 0.7 -fold enrichment inside clusters relative to outside clusters in both nc13 and nc14 in α -amanitin injected embryos (Fig. 4b). To ensure that the observed enrichment inside clusters is not an artifact resulting from performing analysis within restricted regions of a nucleus, we quantified the bound fraction in control regions of the same size as the clusters but located randomly within the nucleus and not overlapping with clusters. In vehicle-injected embryos, the bound fraction inside clusters was 1.3 ± 0.3 -fold higher than in control spots in both nc13 and nc14 (Fig. 4b). The bound fraction within clusters does not increase from nc13 to nc14 in α -amanitin injected embryos, similar to the global population (Fig. 2). In sharp contrast, in vehicle injected embryos, we found a 1.4 ± 0.2 -fold increase in the bound population from nc13 to nc14.”

(ii) The fold-change values now also include the errors as suggested by the reviewer. These errors are calculated using the standard propagation of error.

(iii) We were initially defining 10 control spots for each cluster identified. With each spot having a radius of 200 nanometers, we were only sampling approximately 6% of the nuclear area being imaged. We now increased the number of control spots to 30, and obtained a bound fraction closer to the “outside the cluster” values. We have accordingly updated Fig.4 in the revised manuscript. Additionally, we find that increasing the number of control spots any further does not change the bound fraction, suggesting that ~30 control spots is an appropriate number for this analysis (Fig. R1).

(iv) Finally, to address the reviewer’s comment about the methodology regarding the 6-fold increase in the elongating fraction, we have included a new paragraph in the Methods section (lines 837-850) detailing the decomposition (both for Fig. 2e and for Fig. 4d). We do not make any assumptions here regarding the bound fractions inside vs outside the clusters.

Minor comments:

"mean long-lived cluster lifetime is 6 ± 2 minutes in both nc13 and 7 ± 1 minutes in nc14. "
remove "both"

We have removed this in the main text.

Legend Fig 3(a) and S3.1 should include more information what is displayed. From main text, I assume this is images of PolIII-eGFP?

We have added this information in the figure captions for Fig. 3a, Fig. 3e, and Fig. S3.1.

Title section "RNAPII clusters are composed of elongating and kinetically trapped molecules after ZGA" What does "kinetically trapped molecules" mean? Please rephrase
Fig 6 show elongating Pol II (producing transcripts, positioned after TSS) that are colored as initiating pol II. Please correct.

In this context, "kinetically trapped molecules" imply molecules that show anisotropic exploration, i.e. they have a higher propensity to revisit spaces occupied in previous time points rather than efficiently and isotropically exploring the nucleus. An explanation of this is present in our current text in the same section where we state: "The increased anisotropy in clusters suggests that molecules within them are exhibiting compact exploration kinetics and are likely kinetically confined through frequent rebinding or through an increase in local protein-protein interactions." Based on this description we have updated the title to say "kinetically confined molecules"

We have corrected Fig. 6 based on the Reviewer's note.

Reviewer #4 (Remarks to the Author):

In this study by Apratim et al., the authors use single-molecule tracking and lattice light sheet microscopy to characterize the kinetics of RNAPII before (nc13) and during ZGA (nc14) in *Drosophila*, showing an increase in the bound fraction of RNAPII. By using drugs to inhibit either initiation or elongation, they further dissect the bound fraction into initiating, elongating, and non-specific binding populations, and examine how this affects cluster density and lifetime of RNAPII clusters. They demonstrate an increase in the elongating RNAPII fraction within clusters from nc13 to nc14, and an increase in anisotropy of free RNAPII in clusters. Finally, they show the co-localization of an RNAPII cluster with a hunchback (*hb*) reporter gene, and that the cluster intensity positively correlates with the burst intensity of the *hb* gene.

Overall, this study provides an excellent demonstration of the RNAPII functional states in RNAPII clusters at different developmental stages and opens further questions in the field regarding the molecular control of these cluster dynamics and gene regulation. I believe this study has a solid foundation, and I have no doubt about the model the authors suggest, which is convincing. However, there are several technical areas that require revision:

1. Abstract: "We show that single clusters are stably associated with active gene loci during transcription." The authors only demonstrate this with one gene locus (hb).

We have updated Fig. 5 in the main text to include three more genes: *eve*, *sna* and *sog* in addition to *hb* from our original manuscript. We observe stable associations of RNAPII clusters with all 4 genes during a transcription burst and no stable associations with a non-transcribing control locus (Fig S5.1d).

2. The authors used saSPT for their trajectory analysis and attributed the intermediate state likely as confined molecules. This is a misconception: i) In the standard saSPT pipeline, all array fitting is based on Brownian motion. It is conceptually incorrect to classify molecules with lower fitted diffusion coefficients as confined, although they could be related. Confinement should be justified by anisotropy or other models such as radius of confinement. ii) In their methods, 100 diffusion coefficients were used for fitting. In the original saSPT paper, they emphasized selecting a parameter grid with spacing fine enough to avoid discretization artifacts. How did the authors determine the spacing? And how does changing that parameter affect the intermediate population? That should be further addressed and clarified. iii) The intermediate population could alternatively be explained by freely diffusing populations containing molecular species with different molecular weights. For example, the fast population could be RPB1 alone, while the intermediate could be the full RNAPII complex (or complex with TFIIIF), or the fast population could include truncated protein products. To show whether there are truncated products, Fig. S1.1 shows a Western blot, but it is not a full blot and only uses CTD Ser5P antibody, which is unusual. I think a pan-RPB1 antibody should be used. Fig. S1 also needs quantification to show similar expression levels. Overall, additional experiments and analysis are required to clarify the fast and intermediate populations. iv) For all saSPT plots, it is important to label the y-axis as "posterior probability" to inform readers unfamiliar with the technique that it is a fitted distribution.

We address each point made by the reviewer here in turn:

(i) We agree that it is difficult to definitively classify the intermediate population as confined molecules based solely on saSPT analysis. As the reviewer notes, this population may reflect a mixture of molecular species or behaviors, including molecules that are part of higher-order RNAPII complexes (e.g., with TFIIIF or other general transcription factors), RNAPII species with truncations, or molecules that are transiently confined due to molecular crowding or chromatin interactions. To reflect this, we have revised the Results section (lines 92-96) to state:

“Based on the diffusion coefficient distributions of H2B and NLS, we categorized RNAPII trajectories as (i) chromatin bound; (ii) **intermediate, which may reflect a heterogeneous population of molecules including those transiently confined, diffusing in complex with co-factors, truncated products with different molecular weights**, or (iii) fast, freely diffusing molecules...”

We note that while saSPT analysis does not directly model confinement, we had independently quantified confinement using the anisotropy analysis in Fig. 4c as suggested. These measurements show that molecules within RNAPII clusters exhibit increased confinement compared to both the nucleoplasm and control spots, suggesting that at least a subset of the intermediate state reflects restricted mobility in clusters.

(ii) Regarding the grid spacing in the saSPT fitting: in our analysis, we extended the lower range of diffusion coefficients relative to the original saSPT paper, but kept the number of

diffusion coefficients at 100. To test whether grid spacing impacts the assignment of populations, we repeated the analysis for RPB1-mEos3.2 in nc14 using both half the spacing (200 diffusion coefficients) and double the spacing (50 diffusion coefficients). In both cases, the resulting posterior distributions were nearly identical in shape, and the estimated bound and intermediate fractions were unchanged (Fig. R3). This confirms that our conclusions are not sensitive to discretization artifacts.

Fig. R3: Effect of grid size on fractions. (a) Diffusion coefficient distributions for vehicle-injected embryos in nc14 for three different diffusion coefficient grid sizes. A grid size of 100 was used in the main manuscript. (b) Bound and intermediate (int.) fraction values for the three different grid sizes shown in (a). The data points represent individual fields of view and error bars represent standard deviation. A total of 205,532 trajectories from 10 embryos were obtained for this analysis.

(iii) We agree that the intermediate population could reflect RNAPII species with truncations thus reflecting different molecular weights. To this end, in addition to our previous Western blots with the Ser5P antibody, we also conducted full Western blots using the pan-RPB1 antibody (updated Fig. S1.1). In both cases we observe two bands in the RPB1 blot which has previously been reported as the phosphorylated and unphosphorylated states of the RPB1 CTD¹² as well as some lower molecular weight bands that may be truncated products. To reflect this, we have revised the Results section (lines 92-96) to state:

“Based on the diffusion coefficient distributions of H2B and NLS, we categorized RNAPII trajectories as (i) chromatin bound; (ii) intermediate, which may reflect a heterogeneous population of molecules including those transiently confined, diffusing in complex with co-factors, truncated products with different molecular weights, or (iii) fast, freely diffusing molecules...”

We have also included quantification of the protein concentration levels, to the extent that Western blot analysis permits, from our pan-RPB1 Western blots. The mEos expression appears to be lower in some replicates, but this difference is not statistically significant. Both eGFP and mEos3.2 tagged fly lines and embryos are homozygous viable and have been propagated as such for more than two years with no loss of fecundity.

(iv) We have updated the y axis for all the saSPT plots to state “Posterior probability”.

3. Given the high concentration of RNAPII at the HLB, did the SMT analyses exclude trajectories from the HLB? Fig. 4 seems to exclude HLB data, but what about other SMT data? If not, how might the properties of RNAPII at HLB skew the SMT data at different conditions? This should be clarified.

The overall SMT data (Fig. 1,2) does not currently exclude trajectories from the HLB across any of the conditions (drugs, vehicle). However, to address the reviewer’s concern, we re-analyzed the data after removing trajectories from the putative HLBs identified already in Figure 4. On doing so, we found that the overall diffusion coefficient distributions were identical both in nc13 and nc14 (Fig. R4). This suggests that even after including the HLB data our main conclusions remain unchanged.

Fig. R4: Effect of removing HLB associated trajectories on fractions. (a) Diffusion coefficient distributions for vehicle-injected RPB1-mEos3.2 embryos in nc13 (top) and nc14 (bottom) when considering all the trajectories (blue) and excluding trajectories associated with the histone locus bodies (HLB, purple). **(b)** Bound fraction in both nc13 and nc14 when all trajectories are considered versus when trajectories associated with the HLBs are excluded. A total of 210,638 and 205,532 trajectories were obtained in nc13 and nc14 respectively for the

overall case. For the HLB excluded case, a total of 181,302, and 187,694 for nc13 and nc14 were obtained respectively.

4. When quantifying changes in bound fraction (e.g., Fig. 1d, 2d), it is important to consider changes in the number of RNAPII molecules per cell. In Fig. S2.1d, nc13 appears to have higher nuclear intensity of RNAPII than nc14. If the authors incorporate this into their calculations, how would the amount of bound RNAPII change, and how would that affect their biological interpretation?

The change referred to by the reviewer in Fig. S2.1d reflects a slight decrease in mean nuclear RNAPII intensity from nc13 to nc14. However, this difference was not statistically significant and therefore was not highlighted in the main text. To further address the reviewer’s concern, we also analyzed RNAPII intensity distributions at the single-nucleus level rather than averaged across embryos and did not observe a statistically significant difference between nc13 and nc14 (Fig. R5). Thus, incorporating the changes in nuclear intensity would not alter our overall conclusion that the bound fraction of RNAPII increases from nc13 to nc14. In fact, given that the nuclear intensity is slightly lower in nc14, the observed increase in the bound fraction would represent an even more pronounced increase in the number of bound molecules than would be the case if nuclear intensity were higher in nc14.

Fig. R5: Quantifying nuclear mean intensity across nuclei. Bar plots show the mean nuclear intensity in nc13 and nc14 across different injection conditions. Each data point represents an individual nucleus. n values are 115 and 139 for vehicle injection, 144 and 266 for α-amanitin injection, and 68 and 162 for triptolide injection in nc13 and nc14 respectively. Error bars represent standard deviation.

5. I appreciate the approach of using drugs to dissect initiation, elongation, and non-specific binding. However, further experiments are needed to make the quantification convincing. (i) The claim regarding non-specific binding does not seem robust. As shown in Fig. 3, the drug is not effective at HLB, indicating that the drugs could have gene-specific effects. To claim non-specific binding, the authors should at least conduct a dose-dependent assay to demonstrate they are

using sufficient drug concentrations to inhibit almost all initiation or elongation events (shown by no further change in bound fraction) at non-HLB loci. (ii) Many claims about RNAPII states in clusters across developmental stages should also be validated using CTD Ser2P and Ser5P antibodies in fixed cells, as the drug assays do not provide direct evidence. (iii) Additionally, how do triptolide and α -amanitin affect paused RNAPII? It should be addressed experimentally or at least discussed in the manuscript.

(i) In the manuscript, we used a triptolide concentration of 0.5 mg/ml for embryo injections, We originally validated this concentration by confirming that the MS2-MCP signal marking nascent *hunchback* transcription was abolished within ~5 minutes of injection (Fig. S2.1e).

To directly address the reviewer's concern about the triptolide dosage, we conducted a dose-response assay using triptolide concentrations ranging from 0.5 to 2.5 mg/ml. Embryos tolerated concentrations up to ~1.75 mg/ml with modest effects on viability, while higher concentrations caused immediate developmental arrest. Furthermore, single-molecule tracking (SMT) at the higher concentration of 1.75 mg/ml yielded a bound fraction ($12\pm 4\%$) comparable to that at 0.5 mg/ml ($14\pm 5\%$), indicating that the inhibitory effect of the drug had already saturated at the concentration used in the manuscript. This supports our interpretation that transcription initiation is effectively blocked at a concentration of 0.5 mg/ml and that the remaining bound fraction likely reflects non-specific RNAPII-chromatin interactions. We further confirmed this with EUTP injection at the 0.5 mg/ml concentration of Triptolide which showed nucleus-wide loss of nascent transcripts. This data is now included in the revised Fig. S2.1 b,c.

With regards to histone locus bodies (HLBs), previous work has already demonstrated that these structures are resistant to transcriptional perturbations (lines 246-248 in the manuscript). For instance, Cho et al., 2023 showed that HLBs persist after α -amanitin injection¹³, and Huang et al., 2021, reported their continued presence in embryos lacking *Zelda*, a key pioneer factor necessary for genome activation¹⁴. Consistent with this, excluding trajectories associated with HLBs from our analysis did not alter the calculated bound fraction (Fig. R4). Together, this indicates that the presence of HLBs after the drug injections does not confound our interpretations.

(ii) In line with the reviewer's suggestion, we performed separate Ser2P and Ser5P immunofluorescence staining in eGFP-RPB1 embryos at early stages (nc11-12, prior to ZGA) and in nuclear cycle 14 (after ZGA) to assess for co-localization. We have included the results from this analysis in the new Fig. S3.3. Manders coefficient analysis between the two channels revealed that in the early cycles, the coefficients were significantly higher for Ser5P, whereas in nc14 they were higher for Ser2P. These results support our conclusion that RNAPII clusters are predominantly associated with initiation in the early cycles and shift towards elongation following ZGA.

(iii) We have revised the main text (lines 126-130) to more explicitly discuss the mechanisms of the inhibitors used as shown below :

“To assess the relative contributions of molecules engaged in transcription initiation and elongation to the bound fraction of RNAPII, we injected embryos with either triptolide, which blocks formation of the transcription initiation complex by targeting TFIIF¹⁵, or α -amanitin, which binds to RNAPII directly and slows the elongation rate to an extent that transcription is effectively inhibited^{16,17} (Fig. 2a).”

Thus, while triptolide is expected to indirectly reduce paused polymerases by disrupting transcription initiation, α -amanitin is expected to increase the proportion of polymerases that remain promoter-proximal and non-productive by slowing elongation.

6. Regarding Fig. 3e, can the authors explain why α -amanitin causes a decrease in cluster density while the remaining clusters seem to be becoming larger? Do the clusters fuse? Can the authors examine the differences in the size of the clusters in different conditions for Fig. 3e?

Since α -amanitin disrupts transcription elongation, we observed a reduction in the overall number of clusters, with the remaining clusters likely corresponding to transcription initiation events. Importantly, we did not observe any cluster fusion events.

We also did not observe any significant difference in the cluster sizes after injecting either α -amanitin or Triptolide, as shown in Fig. R6. These findings suggest that cluster size is not directly modulated by transcriptional activity but rather intensity is reflecting the number of engaged polymerases.

Fig. R6: Bar plots showing the average cluster sizes across the different conditions. n=5, 3 for the control embryos in nc 13, 14 respectively, n=3 for both triptolide and α -amanitin injected embryos in nc 13,14. Error bars represent standard deviation.

7. The authors show that α -amanitin increases long-lived cluster lifetime in Fig. 3g,h. I would expect to see that reflected in Fig. 4a, but this does not appear to be the case. Presumably, if the "bound trajectories" were longer or appeared more frequently in the clusters, they should be prominent in the trajectory plot in Fig. 4a.

Although α -amanitin increases cluster lifetimes (Fig. 3), this effect is not reflected as a higher density of bound trajectories in the spatial trajectory maps (Fig. 4). This

discrepancy likely arises because the mechanisms underlying the longer cluster lifetimes are multifactorial. First, our global SMT analysis (Fig. 1) shows that α -amanitin increases the intermediate RNAPII population. This is consistent with previous studies which show that α -amanitin traps RNAPII in an open, inactive conformation by interfering with the trigger loop and bridge helix, causing it to cycle between initiation and premature eviction from chromatin^{16,18,19}. This is discussed in the manuscript in lines 178-181:

“This is consistent with previous studies which suggest that α -amanitin traps RNAPII in an open inactive conformation by interfering with the trigger loop and bridge helix, causing it to cycle between initiation and premature eviction from chromatin^{16,18,19}.”

Second, our anisotropy analysis (Fig. 4), which excludes bound molecules, shows higher confinement of trajectories within clusters in α -amanitin-injected embryos in nc14, consistent with more restricted molecular motion. Together, these results suggest that the increased cluster lifetime arises not from a higher fraction of bound trajectories, but rather from an increased fraction of molecules in intermediate states and enhanced kinetic trapping within clusters, perhaps through rapid re-engagement at the promoter. We have added this to the Discussion section (lines 390-395):

“In nc14, we also observe an increase in the anisotropy of non-bound RNAPII molecules within clusters, which suggests a kinetic confinement of molecules near active transcription sites (Fig. 4). This confinement becomes more pronounced upon transcription inhibition with α -amanitin, suggesting that the increased recycling of RNAPII molecules at promoters can extend cluster lifetimes without altering the overall bound fraction^{16,18}.”

8. Regarding anisotropy: From Fig. S4.2, H2B trajectories in nc14 were used to determine the distance threshold. (i) Can the authors also do the same analysis with H2B trajectories in nc13? If the two developmental stages exhibit distinct transcriptional activity, I would expect chromatin dynamics and H2B mobility to differ, potentially requiring different distance thresholds. (ii) Second, if I understand correctly, for Fig. 4, the "RNAPII cluster" is not a cluster at a given time but is defined by the average position of trajectories over an 80-second period. This means that if RNAPII clusters move during this period (which they certainly do, as shown in Fig. 3 and Fig. S3.1 on a 10-second scale), then the analysis area represents the space explored by RNAPII clusters over time, not at a specific moment. If cluster mobility differs between nc13, nc14, or after α -amanitin treatment, the anisotropy values would be confounded. I understand that the SMT was done at a very fast time scale (10ms/frame), but such movement of clusters should not be overlooked. This technical limitation could explain the modest 1.3-fold change, which might not be biologically meaningful.

(i) The reviewer is correct that we initially used the step size distribution from H2B trajectories in nc14 to define the anisotropy threshold for both nc13 and nc14. To address this, we have now included the step size distribution from nc13 in the revised Fig. S4.1g. The distributions for nc13 and nc14 are highly similar in both shape and range, and the

0.2 μm cutoff remains near the 95th percentile in each case. Based on this close agreement, we consider it appropriate to use a common 0.2 μm threshold for anisotropy calculations across both developmental stages.

(ii) While it is true that the anisotropy measurements are made within regions defined by time-average RNAPII cluster positions (~ 60 sec), the relevant time scale for the anisotropy analysis itself is orders of magnitude shorter. Individual trajectories contributing to anisotropy measurements span ~ 2 -15 frames (i.e. 20-150 msec). In contrast, our volumetric imaging (10 sec/frame) reveals that RNAPII clusters show minimal movement between two consecutive frames (i.e across 10 sec) as shown in the representative example in Fig. R7. Given this large temporal separation, the angular information extracted from individual SMT trajectories remains valid and is unlikely to be confounded by bulk cluster motion over time. This clarification has been added to the Methods section in lines 885-887:

“Furthermore, the timescale of anisotropy measurements (~ 20 -150 msec from individual trajectories) is much shorter than the timescale of bulk cluster motion which occurs over tens of seconds.”

Fig R7: RNAPII clusters shown 10 seconds apart (two consecutive time points) and the associated merged image.

9. Line 252 reports a "6-fold increase in the elongating population of RNAPII within clusters from nc13 to nc14" which is a key result to me. It would be helpful to show the underlying calculation, perhaps in a format similar to Fig. 2e, to clarify how this value was derived.

We have now added a new plot showing the increase in the elongating fraction (Fig. 4d). We have also included details about the underlying calculation in the Methods section.

Bibliography

1. Cho, C.-Y., Kemp, J. P., Jr, Duronio, R. J. & O'Farrell, P. H. Coordinating transcription and replication to mitigate their conflicts in early *Drosophila* embryos. *Cell Rep* **41**, 111507 (2022).

2. Stavreva, D. A. *et al.* Transcriptional Bursting and Co-bursting Regulation by Steroid Hormone Release Pattern and Transcription Factor Mobility. *Mol Cell* **75**, 1161–1177.e11 (2019).
3. Hug, C. B., Grimaldi, A. G., Kruse, K. & Vaquerizas, J. M. Chromatin Architecture Emerges during Zygotic Genome Activation Independent of Transcription. *Cell* **169**, 216–228.e19 (2017).
4. Darzacq, X. *et al.* In vivo dynamics of RNA polymerase II transcription. *Nat Struct Mol Biol* **14**, 796–806 (2007).
5. Kwasnieski, J. C., Orr-Weaver, T. L. & Bartel, D. P. Early genome activation in is extensive with an initial tendency for aborted transcripts and retained introns. *Genome Res* **29**, 1188–1197 (2019).
6. Saunders, A., Core, L. J., Sutcliffe, C., Lis, J. T. & Ashe, H. L. Extensive polymerase pausing during *Drosophila* axis patterning enables high-level and pliable transcription. *Genes Dev* **27**, 1146–1158 (2013).
7. Lott, S. E. *et al.* Noncanonical compensation of zygotic X transcription in early *Drosophila melanogaster* development revealed through single-embryo RNA-seq. *PLoS Biol* **9**, e1000590 (2011).
8. Ciabrelli, F., Atinbayeva, N., Pane, A. & Iovino, N. Epigenetic inheritance and gene expression regulation in early *Drosophila* embryos. *EMBO Rep* **25**, 4131–4152 (2024).
9. Chen, K. *et al.* A global change in RNA polymerase II pausing during the *Drosophila* midblastula transition. *Elife* **2**, e00861 (2013).
10. Henninger, J. E. *et al.* RNA-Mediated Feedback Control of Transcriptional Condensates. *Cell* **184**, 207–225.e24 (2021).
11. Gaskill, M. M. *et al.* Localization of the *Drosophila* pioneer factor GAF to subnuclear foci is driven by DNA binding and required to silence satellite repeat expression. *Dev Cell* **58**, 1610–1624.e8 (2023).

12. Website. Anti-RNA polymerase II RPB1 (phospho S5) antibody.
<https://www.abcam.com/en-us/products/primary-antibodies/rna-polymerase-ii-rpb1-phospho-s5-antibody-ab240740#>.
13. Cho, C.-Y. & O'Farrell, P. H. Stepwise modifications of transcriptional hubs link pioneer factor activity to a burst of transcription. *Nat Commun* **14**, 4848 (2023).
14. Huang, S.-K., Whitney, P. H., Dutta, S., Shvartsman, S. Y. & Rushlow, C. A. Spatial organization of transcribing loci during early genome activation in *Drosophila*. *Curr Biol* **31**, 5102–5110.e5 (2021).
15. Chen, F., Gao, X. & Shilatifard, A. Stably paused genes revealed through inhibition of transcription initiation by the TFIIH inhibitor triptolide. *Genes Dev* **29**, 39–47 (2015).
16. Brueckner, F. & Cramer, P. Structural basis of transcription inhibition by alpha-amanitin and implications for RNA polymerase II translocation. *Nat Struct Mol Biol* **15**, 811–818 (2008).
17. Rudd, M. D. & Luse, D. S. Amanitin greatly reduces the rate of transcription by RNA polymerase II ternary complexes but fails to inhibit some transcript cleavage modes. *J Biol Chem* **271**, 21549–21558 (1996).
18. Kaplan, C. D., Larsson, K.-M. & Kornberg, R. D. The RNA polymerase II trigger loop functions in substrate selection and is directly targeted by alpha-amanitin. *Mol Cell* **30**, 547–556 (2008).
19. Xu, L. *et al.* Dissecting the chemical interactions and substrate structural signatures governing RNA polymerase II trigger loop closure by synthetic nucleic acid analogues. *Nucleic Acids Res* **42**, 5863–5870 (2014).

Revision document

We thank the reviewers for their thorough review of our resubmitted manuscript and associated responses to their comments, and their broad support for publication of this manuscript pending some minor points. **Point-by-point responses and changes are detailed below in bold text.**

Reviewer #1::

I would like to thank the authors for the thorough review of their manuscript. All my comments are now satisfactorily addressed.(Remarks on code availability)

We thank the reviewer for their contributions to the review and for supporting the publication of this manuscript.

Reviewer #3:

The revision of Mukherjee et al. is much improved and I appreciate the rephrasing and additional analysis the authors have performed. I only have two remaining minor comments. If properly addressed, I fully support publication of this manuscript in Nature Communications.

1. The authors addressed my comments of Fig 2e only partially. I appreciate the changes the authors made to this figure and it helps to have the nc13 and ncn14 next to each other for comparison. However, I still think it is confusing to normalize to the total bound Pol II, as this is changing between the conditions. This also does not match the description in the text: "This decomposition shows that the elongating fraction of RNAPII increases by $\sim 1.8 \pm 0.5$ -fold from nc13 to nc14", whereas figure 2e only shows a fold change of 1.2, from 0.45 to 0.55. Can the authors normalize this fraction to the total pol II, so the fraction represents the actual fraction of Pol II in the cells (rather than a fraction of a bound fraction)?

We have updated Fig. 2e in the main manuscript by removing the normalization to the total bound RNAPII. In our updated figure we simply show the decomposition of the bound fraction calculated in Fig. 2d for nc13 and nc14. This decomposition, without the normalization, clearly shows the ~ 1.8 -fold increase in the elongating population during ZGA. Furthermore, in order to remain consistent with our data display, we have similarly updated Fig. 4d to remove the normalization to the total bound population.

2. In several instances, the authors mention "nascent transcription". Since transcription is always nascent, I suggest the authors use either "nascent transcripts" or "transcription".

We have updated all instances in the main text and figure captions as per the reviewer's recommendation here.

Reviewer #4

The authors have done substantial work to address my concerns. The addition of analysis for three more genes significantly strengthens the study. Overall, the main conclusions are justified. However, I have three remaining points that require clarification. I do not request additional experiments, but the authors should address the following issues:

1. Regarding the statement "intermediate, which may reflect a heterogeneous population of molecules including those transiently confined, diffusing in complex with co-factors, truncated products with different molecular weights": In my original comment, I meant that the FAST population might include truncated products, not the intermediate population, as truncated products are expected to diffuse with a larger, not a smaller, diffusion coefficient.

We thank the reviewer for providing additional clarification. We have now updated our manuscript to read as follows:

Lines 93-96: Based on the diffusion coefficient distributions of H2B and NLS, we categorized RNAPII trajectories as (i) chromatin bound; (ii) intermediate, which may reflect a heterogeneous population of molecules including those transiently confined, or diffusing in complex with co-factors, or (iii) fast, which may reflect freely diffusing molecules or truncated products with reduced molecular weights.

2. Fig. R3 appears to contain errors. The authors state 200 diffusion coefficients, which I assume equals 200 grid spacings. However, Fig. R3 indicates 150 for grid spacing. The authors also mention this is from nc14, but by comparing with Fig. 1c, the data appears to be from nc13 instead. Please clarify.

We apologize for the oversight regarding Fig R3. We had stated the 200 grid size in the text but had plotted the curve for the 150 grid size. Additionally, the plot was for nc13 as the reviewer rightly points out. To address these issues, we now show the grid sizes for 50, 100 (used in the manuscript), 150, and 200 for both nc13 and nc14 in the new figure R1 below. As demonstrated in the earlier revision, adjusting the grid sizes does not change the calculated bound or intermediate fraction.

Fig. R1: Effect of grid size on fractions. (a) Diffusion coefficient distributions for vehicle-injected embryos in nc13 (left) and nc14 (right) for four different diffusion coefficient grid sizes. A grid size of 100 was used in the main manuscript. (b) Bound and intermediate (int.) fraction values for the

four different grid sizes shown in **(a)**. The data points represent individual fields of view and error bars represent standard deviation. A total of 210,638 trajectories from 9 embryos were obtained in nc13, and 205,532 trajectories from 10 embryos in nc14.

3. The pan-RPB1 Western blot in Fig. S1.1 raises concerns about data interpretation. The error bars are quite large, with 0.4-fold differences observed for some replicates in panel e. The representative blot shown in panel d appears much stronger in eGFP-RPB1 than the quantified average suggests. The lack of statistical significance in panel e does not necessarily mean the samples are equivalent or similar to the untagged control, but rather reflects high variability in the data. To provide transparency, please include all four replicate Western blots in the supplement. Additionally, please clarify whether the quantification in panel e is based on the pan-RPB1 antibody or the Ser5 antibody.

We now include all Western blot replicates for the pan RPB1-antibody as the reviewer suggests. We have also added language in the figure caption to clarify that the quantification in panel e is based on the pan RPB1-antibody. Specifically, we say “(e) Quantification of protein concentration normalized to the OreR across the Western blot replicates for the pan-RPB1 antibody antibody shown in (d)”.

We agree that the pan-RPB1 blots show high variability, which we attributed to technical difficulties and a previously untested antibody, which is why we initially used the Ser5 antibody. To reduce potential sources of technical error we repeated the Western blots for the pan-RPB1 antibody using the Jess automated Western blot system. We include blots for three additional replicates from this automated analysis in figure S1.1 new panel f and associated quantification in new panel g. This new quantification yields similar results as before (panel e), showing higher variability than one might expect from RPB1 but no significant difference between protein concentration normalized to OreR for the GFP and mEos3.2 constructs. We do agree with the reviewer that there is a relatively higher variability in the expression level for the GFP tag compared to the mEos3.2 tag (both from our earlier analysis and the new Jess analysis). We suspect the source of this variability is technical in nature and it is unlikely that it reflects changes in RNAPII nuclear concentrations (embryo to embryo) as assessed by fluorescence analysis (Fig S2.1g).